# Programmed cell death regulator BAP2 is required for IRE1-mediated unfolded protein response in Arabidopsis

Noelia Pastor-Cantizano [1,5], Evan R. Angelos[1,6], Cristina Ruberti [1,7], Tao Jiang[1,8], Xiaoyu Weng[2], Brandon C. Reagan[1], Taslima Haque[2,9], Thomas E. Juenger [2] & Federica Brandizzi [1,3,4] ✉

Environmental and physiological situations can challenge the balance between protein synthesis and folding capacity of the endoplasmic reticulum (ER) and cause ER stress, a potentially lethal condition. The unfolded protein response (UPR) restores ER homeostasis or actuates programmed cell death (PCD) when ER stress is unresolved. The cell fate determination mechanisms of the UPR are not well understood, especially in plants. Here, we integrate genetics and ER stress profiling with natural variation and quantitative trait locus analysis of 350 natural accessions of the model species *Arabidopsis thaliana*. Our analyses implicate a single nucleotide polymorphism to the loss of function of the general PCD regulator *BON-ASSOCIATED PROTEIN2* (BAP2) in UPR outcomes. We establish that ER stress-induced *BAP2* expression is antagonistically regulated by the UPR master regulator, inositol-requiring enzyme 1 (IRE1), and that *BAP2* controls adaptive UPR amplitude in ER stress and ignites pro-death mechanisms in conditions of UPR insufficiency.

ER stress is detected through specialized molecular sensors associated with the ER membrane that define different arms of the UPR[1,2]. The most conserved sensor in eukaryotes is the inositol-requiring enzyme 1 (IRE1), a bifunctional kinase and ribonuclease that catalyzes the unconventional splicing of a specific basic leucine zipper (bZIP)-transcription factors (TF) mRNA[3,4]: *HAC1* in yeast[5,6], *XBP1* in metazoans[7,8] and *bZIP60* in plants[9–11]. The spliced mRNA encodes an active TF that is translocated into the nucleus for the transcriptional regulation of UPR target genes. In plants and metazoans, the UPR has evolved additional UPR transducers, such as the ER membrane-tethered bZIP TF ATF6 in mammals and the plant functional homologs bZIP17/ bZIP28[12,13]. Under ER stress, ATF6 and bZIP28 are proteolytically activated in the Golgi for the release of the active TF domain, which is translocated into the

nucleus to modulate UPR gene expression[12,14,15]. Metazoans have developed a third arm of the UPR, which is defined by the ER membrane protein kinase RNA-like ER kinase (PERK)[16].

The model plant species *Arabidopsis thaliana* genome encodes two functional IRE1 paralogs, IRE1A and IRE1B, which closely resemble the yeast and mammalian IRE1[17]. Under chronic ER stress conditions, the loss of both IRE1A and IRE1B or of both bZIP28 and bZIP60 accelerates the progression to cell death; however, the loss of either bZIP60 or bZIP28 is viable[11,15,18,19]. Hence IRE1 and bZIP28/60 have a protective role and function to antagonize endogenous cell death processes that are actuated in conditions of chronic ER stress.

In the absence of induced ER stress, the loss of a functional IRE1 causes growth defects in Arabidopsis, specifically root growth[18], and

[1]MSU-DOE Plant Research Lab, Michigan State University, East Lansing, MI, USA. [2]Department of Integrative Biology, University of Texas, Austin, TX, USA. [3]Plant Biology Department, Michigan State University, East Lansing, MI, USA. [4]Great Lakes Bioenergy Research Center, Michigan State University, East Lansing, MI, USA. [5]Present address: Department of Biochemistry and Molecular Biology, Institute for Biotechnology and Biomedicine (BIOTECMED), University of Valencia, Burjassot, Spain. [6]Present address: Botany & Plant Sciences Department, Institute for Integrative Genome Biology, University of California Riverside, Riverside, CA, USA. [7]Present address: Department of Biosciences, University of Milan, Milano, Italy. [8]Present address: Mid-Florida Research and Education Center, University of Florida, Apopka, FL, USA. [9]Present address: Department of Ecology and Evolutionary Biology, University of Michigan, Ann Arbor, MI, USA. ✉e-mail: fb@msu.edu

embryo lethality in mouse[20], indicating that IRE1 is necessary to control processes that favor growth in the absence of induced ER stress. Therefore, the nature of IRE1 activity is very diverse and determined by tissue, physiological state of the cell, stress intensity, and regulatory proteins that are associated with IRE1[21,22]. As such, IRE1 integrates and responds to stress cues in the ER lumen and ER membrane[23–27], as well as soluble and membrane proteins that functionally associate with IRE1[28].

During transient stress, IRE1 is transiently activated and eventually inactivated; however, in chronic stress, its activity is sustained for longer periods, triggering cell death[1,2,22,29]. Therefore, the role of IRE1 in ER stress is biphasic: it directs pro-life activities while monitoring endogenous pro-death processes when ER stress is still resolvable but ignites pro-death processes when ER stress cannot be resolved due to UPR insufficiency. Hence, coordination between adaptive and pro-death responses by IRE1 is crucial for UPR sufficiency in cell fate determination during ER stress. However, the underlying mechanisms are still not well established, especially in plants.

Due to the essential role of IRE1 in cell fate determination, IRE1 signaling amplitude and duration must be tightly regulated. Although in mammalian cells, several co-factors have been reported to interact with IRE1 to regulate IRE1 signaling outputs[21,22], in plants, similar co-factors are still largely unknown. For example, the mammalian ER-membrane protein BAX inhibitor 1 (BI-1) interacts with the C-terminal region of IRE1 to negatively modulate IRE1-endonuclease activity and splicing of *XBP1*, attenuating IRE1's pro-survival role during ER stress adaptation[30]. Conversely, *Arabidopsis* BI1 does not reduce IRE1 splicing activity[31], supporting that plants have evolved unique strategies to control IRE1 activity that are yet unknown.

In animal cells, the mechanisms that lead to apoptosis and programmed cell death (PCD) during ER stress have been extensively investigated, and include conserved components of PCD, such as anti-apoptotic and pro-apoptotic Bcl-2 family members, pro-apoptotic Bcl-2-Associated X/Bcl-2-Antagonist Killer (BAX/BAK) proteins and caspase[21,22]. In plants, two proteases with caspase-3-like activity, PBA1, and cathepsin B, antagonistically regulate PCD during ER stress[32]. Although homologs of many of the animal PCD elements are not present in plants, and conserved elements have been shown not to have a large effect on ER stress survival outcomes[31], plants have developed specific PCD regulators, such as the NAC transcription factors[33]. Within the ER-membrane tethered NAC family members, under ER stress, NAC089 is transcriptionally regulated by bZIP60 and bZIP28, and it has been reported to functionally connect these ER stress sensors and downstream PCD-associated genes upon a proteolytic activation[34].

In this work, we combine genetics and quantitative genomics by leveraging the natural variation of *A. thaliana*, and establish BAP2, a general PCD regulator in plants[35], as a cellular rheostat, which, in ER stress, monitors the sufficiency of the IRE1-bZIP60 UPR arm and acts as a pro-death effector in conditions of UPR insufficiency.

## Results

### *A. thaliana* has extensive natural variation in ER stress response

Understanding natural variation at systems levels is a powerful approach to identifying the underlying mechanisms of complex biological pathways in plants[36]. In humans, natural variants of the ER stress response have been associated with several diseases, including diabetes, amyotrophic lateral sclerosis, and bowl inflammatory disease[37–39]. Therefore, we hypothesized that the plant UPR would exhibit natural variation and that this could be used as a resource to identify modulators of ER stress responses. To test these hypotheses, we analyzed the sensitivity to chronic ER stress in 350 natural *A. thaliana* accessions (Fig. 1a and Supplementary Data 1). We used the customary approach to evaluate plant biomass accumulation in response to ER stress by measuring the shoot fresh weight of seedlings

directly germinated on chronic ER stress-inducing growth media[11,15,18]. Each accession was grown on solid media containing either tunicamycin (TM, a widely used ER stress inducer) or DMSO (TM solvent control, mock) for 10 days. By using linear mixed model analyses testing for genetic variation among accessions, and by estimating the impact of the TM treatment on biomass accumulation compared to the respective mock control, we detected a highly significant accession-by-treatment interaction ($P < 0.001$), highlighting natural variation in the accumulation of biomass of the accessions in response to chronic ER stress. We then visualized and ranked the accessions by the relative biomass ratio (RR) based on fresh weight for each accession relative to the reference accession, Columbia-0 (Col-0), in the presence and absence of TM. The 350 accessions showed a 10-fold range of natural variation in ER stress sensitivity, providing ample genetic resources to analyze the variability to ER stress in intact individuals (Fig. 1a and Supplementary Data 1), in accordance with our initial hypothesis.

Across all the analyzed accessions, *Nantes* (Na-1) showed the highest relative growth ratio (RR) value compared to Col-0 (RR = $1.73 \pm 0.13$), while *Eastland* (Est-0) showed one of the lowest values (RR = $0.21 \pm 0.03$) (Fig. 1a and Supplementary Data 1). These accessions differed by 4.6-fold in TM-induced ER stress sensitivity when they were grown on media containing either 25 or 50 ng mL$^{-1}$ TM (Fig. 1b, c and Supplementary Fig. 1). To test if these two accessions also differed in the transcriptional modulation of UPR target genes, a hallmark of UPR activation[2], we analyzed the transcript levels of *spliced bZIP60* (*sbZIP60*), *Binding Protein 3* (*BiP3*) and *Endoplasmic Reticulum dnaJ domain-containing protein 3B* (*ERdj3B*). *BiP3* is the most representative biomarker indicator of UPR activation[18,40,41], and its induction mainly depends on IRE1-bZIP60, while *ERdj3B* transcriptional activation primarily relies on bZIP28[31]. By using quantitative RT-PCR (qRT-PCR) analyses in seedlings treated with TM for 3 or 6 h, we found marked differences in gene transcript levels between the accessions and Col-0 (Fig. 1d). Compared to Col-0, Est-0 showed higher expression levels of *BiP3* and *sbZIP60* at both TM treatment times, while Na-1 showed similar levels at 3 h, but significantly higher levels of *BiP3* and *sbZIP60* transcripts at 6 h. In contrast, we only found differences in *ERdj3B* expression levels in Est-0 after 6 h of TM treatment. These results indicate the existence of a natural variation in plant UPR activation, which seems to be predominant in the UPR genes mainly controlled by the IRE1-bZIP60 arm in both accessions.

### QTL-seq identifies one genomic region associated with variation in ER stress response

To identify the genetic polymorphism associated with the verified variations in biomass accumulation and molecular signature in response to ER stress in Na-1 and Est-0, we conducted a QTL-seq strategy, combining with bulked-segregant analysis and high-throughput re-sequencing, on a F$_3$ population descendent from a Na-1 x Est-0 cross. To do so, we estimated the RR of 400 F$_3$ progenies using the same method as our screening of the 350 accessions panel (Fig. 2a and Supplementary Data 2). The 10% tail of progeny (42 individuals) showing the most extreme response values were identified as either hyper-sensitive (HP) (lower RR: $0.258 \pm 0.09$ to $0.49 \pm 0.04$) or hyper-resistant (LP) (higher RR: RR = $1.133 \pm 0.07$ to $1.50 \pm 0.2$) populations, respectively. We then extracted DNA from these individuals, pooled them into either a hyper-sensitive or hyper-resistant pool, and sequenced the two bulk populations to a depth of ~ 74 × coverage. We calculated a SNP-index, which is the proportion of SNPs that are different from the Col-0 sequence, for the hyper-resistant (LP) and the hyper-sensitive (HP) pools. Then, we subtracted the SNP-index from the two bulks to obtain another parameter, ΔSNP-index. The average distributions of the SNP-index and ΔSNP-index were estimated in a given genomic interval by a sliding window analysis with 1 Mb window

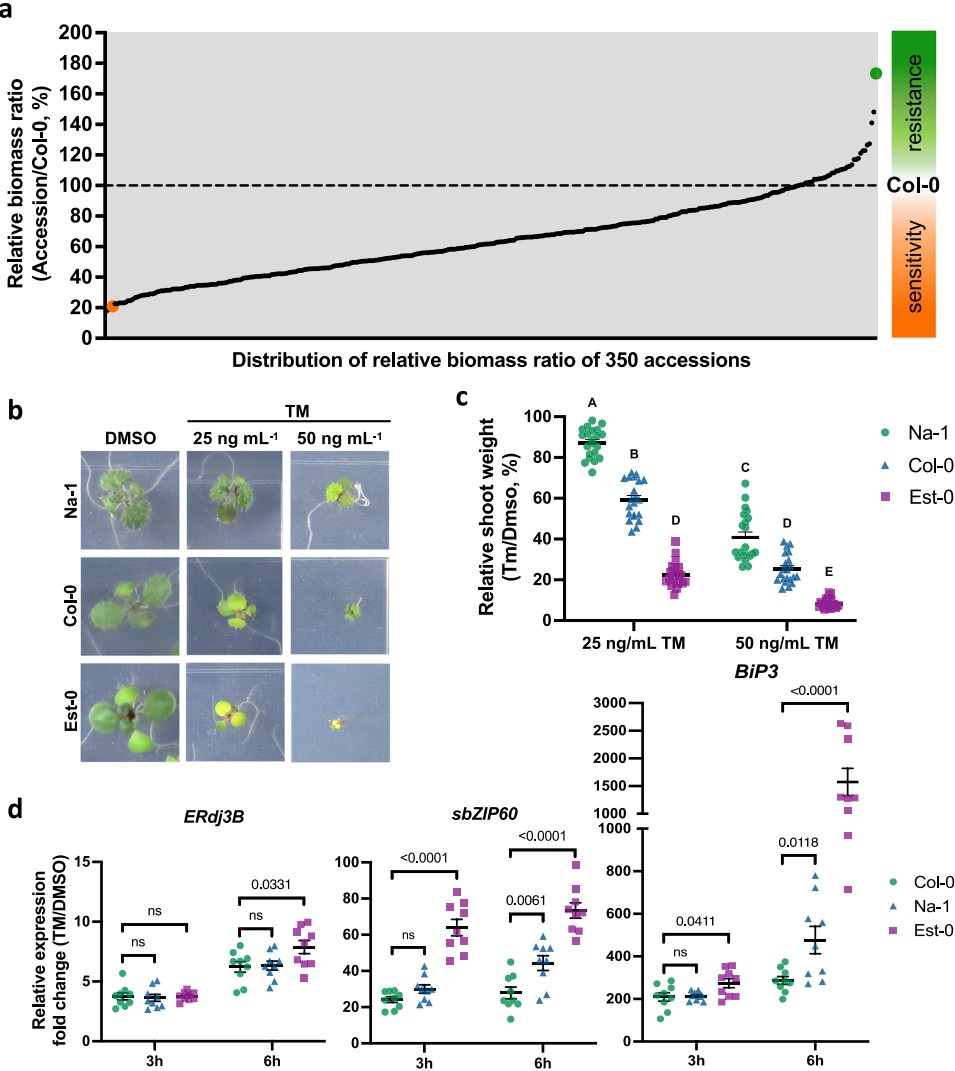

**Fig. 1 | *A. thaliana* shows an extensive natural variation to ER stress.**
**a** Distribution of 350 accessions based on the biomass (shoot fresh weight) ratio of TM treatment vs. untreated (DMSO) compared to the accession reference, Col-0, for which the ratio equals 1 (green and orange dots indicate Na-1 and Est-0 values, respectively). The RR for each accession is given in Supplementary Data 1. **b** Na-1 and Est-0 accessions selected for the QTL mapping analysis were germinated on either 25 or 50 ng mL⁻¹ TM or DMSO for 10 days. **c** Relative shoot weight of Col-0 (green triangles), Na-1 (blue dots), and Est-0 (purple squares) seedlings germinated on media containing the indicated concentrations of TM or DMSO grown for 10 days. Data represent means ± SEM among biological replicates (*n* = 21). Statistical significance was determined using a factorial linear mixed model framework followed by post-hoc testing using the two-sided Tukey's HSD test (multiple testing-controlled threshold used was *P* < 0.05). **d** UPR gene expression analysis in Col-0 (green dots), Na-1 (blue triangles), and Est-0 (purple squares) seedlings after 3- h or 6 h pulse treatment with 0.5 µg mL⁻¹ TM or DMSO. qRT-PCR analyses were performed using specific primers for *ERdj3B*, spliced *bZIP60* (*sbZIP60)* or *BiP3*. No differences were made to the sequence of primers used for the amplification of Est-0, Na-1, and Col-0 accessions based on the 1001 Arabidopsis genome database[56]. *GAPDH* transcript levels were used as an internal control. Gene expression values are presented relative to DMSO control and represent the mean ratio ± SEM among biological replicates (*n* = 9). Statistical significance was determined by Student's unpaired two-tailed *t* test (*P*-value is shown in the graph; ns, not significant). Source data are provided as a Source Data file.

size and 10 kb step (Fig. 2b and Supplementary Data 3). This analysis led to the identification of one genomic region contributing to ER stress sensitivity at the end of chromosome 2 (Chr 2; 95 % confidence interval) (Fig. 2c).

From the 70 protein-coding genes within the chromosomal confidence interval identified in Chr 2 (Supplementary Data 4), we selected candidate genes with known function as a starting point for further analyses. We identified genes involved in stress or physiological responses: *SIN3 ASSOCIATED POLYPEPTIDE 18* (*SAP18*; AT2G45640; salt stress)[42], *GENERAL REGULATORY FACTOR 9* (*GRF9*; AT2G45480; water stress)[43], *SMALL AUXIN UPREGULATED 36* (SAUR36; AT2G45210, leaf senescence and response to auxin and gibberellin[44,45]), *BRI1-5 ENHANCED 1* (*BEN1*; AT2G45400; brassinosteroid response[46]). Furthermore, we selected *CONSTITUTIVELY STRESSED 1* (*COST1*;

AT2G45260) and *BON-ASSOCIATED PROTEIN2* (*BAP2*; AT2G45760), which are involved in autophagy[47] and PCD[35], respectively. We next sought to refine our list of candidates by identifying the genes that transcriptionally respond to ER stress. To do so, we analyzed the transcript levels of these six genes in Col-0 seedlings treated with TM, using qRT-PCR. We utilized an extended time course to cover the adaptive phase of ER stress and a prolonged ER stress condition, and analyzed seedlings at 6, 24, or 48 h of TM treatment (Fig. 2d and Supplementary Fig. 2a). As a control for UPR activation for this time course of ER stress, we analyzed *BiP3* transcript levels (Supplementary Fig. 2b). Except for *BAP2* expression which was increasingly upregulated over the time course of ER stress, the expression of the other selected genes was unchanged. These results led us to focus on *BAP2* as our main candidate for downstream analyses.

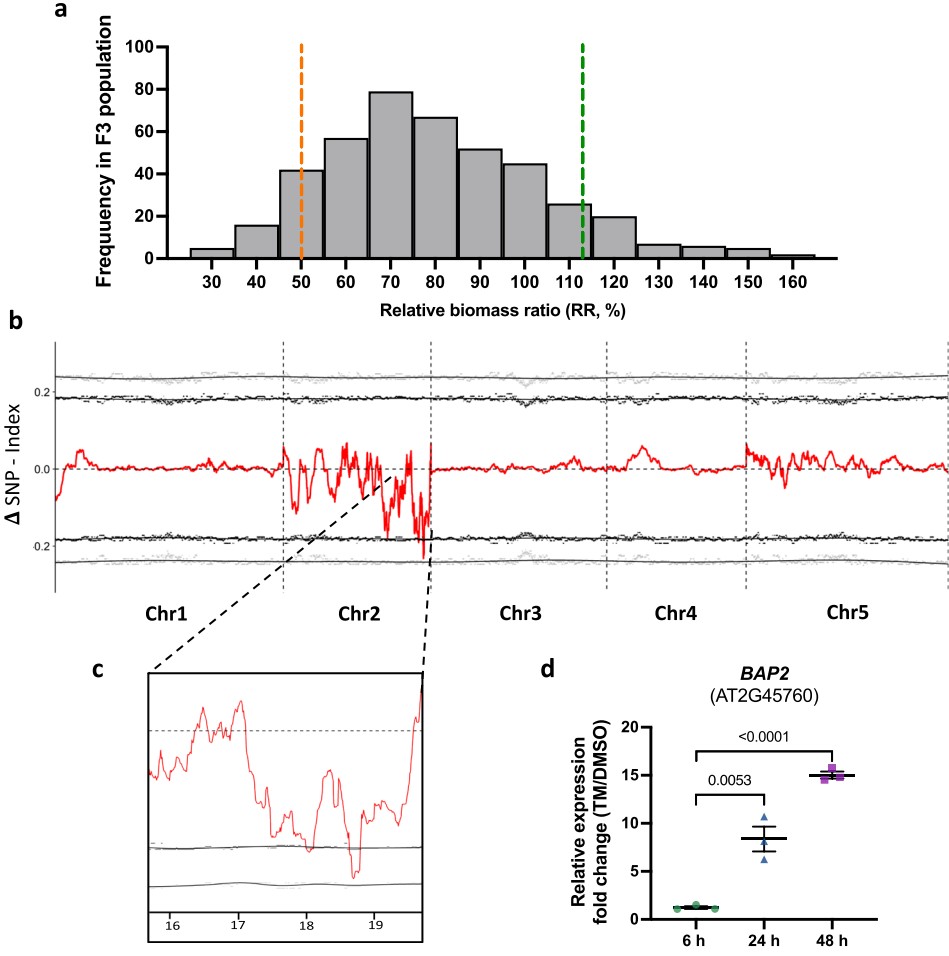

**Fig. 2 | QTL mapping of differential ER stress sensitivity in *A. thaliana* using its extensive natural variation. a** Distribution of the RR of the 400 F$_3$ lines. Orange and green lines represent the cut-off values for the 10% F$_3$ lines considered hyper-sensitive or hyper-resistant populations, respectively. The RR for each accession is given in Supplementary Data 2. **b** Graphical representation of the results of QTL-seq analysis of ER stress sensitivity in *A. thaliana*. The plot represents the allele frequency shifts due to pooling and resequencing of identified hyper-sensitive or hyper-resistant progeny. The red line represents the change in SNP frequency between pools (ΔSNP index), while the black and gray dotted lines are the 95% and 99% confidence intervals of the statistic, respectively (ΔSNP index is

provided in Supplementary Data 3). **c** An expanded graphic of the ΔSNP index plot for the outlier genomic region identified on Chromosome 2. **d** *BAP2* transcription levels analyzed in Col-0 (control accession) upon TM treatment. 5-day-old Col-0 seedlings were grown for 6 (green dots), 24 (blue triangles), and 48 h (purple squares) in plates containing 1 μg mL$^{-1}$ TM or DMSO, and their transcription levels were analyzed by qRT-PCR. *UBQ10* expression levels were used as internal control. Values are presented relative to DMSO control and represent the mean ratio ± SEM among biological replicates ($n = 3$). Statistical significance was determined by Student's unpaired two-tailed *t* test (*P*-value is shown in figure; ns, not significant). Source data are provided as a Source Data file.

## The loss of BAP2 increases sensitivity to chronic ER stress

*BAP2* encodes a small protein (MW: 23 kDa) that contains a calcium-dependent phospholipid-binding C2 domain (Fig. 3c) and is believed to function as a regulator of PCD in plants[35]. To test the role of BAP2 in ER stress resistance as pinpointed in our QTL and gene expression analyses, we tested whether *BAP2* is involved in chronic ER stress. To do so, we analyzed the TM-induced chronic ER stress sensitivity of an established *bap2* knockout mutant (Col-0 background)[35]. Specifically, we estimated the ratio of shoot fresh weight and chlorophyll content of seedlings germinated for 10 days on media containing 15, 25, or 50 ng mL$^{-1}$ TM vs mock-treatment (Fig. 3a, b and Supplementary Figs. 3a, 4a, 5a). As a positive control, we used an established IRE1 mutant (*ire1a ire1b*), which is hypersensitive to chronic ER stress due to UPR insufficiency[11,18]. Consistently, *ire1a ire1b* showed a strong reduction in shoot fresh weight and chlorophyll content on TM-containing growth media (Fig. 3b and Supplementary Figs. 3a, 4a, 5a). Interestingly, we found that *bap2* exhibited a significant reduction in shoot fresh weight and chlorophyll content compared to Col-0. To confirm the sensitivity of *bap2* to ER stress, we analyzed the shoot fresh weight of Col-0, *bap2*, and *ire1a ire1b* seedlings grown on media

containing 1, 2, or 3 mM dithiothreitol (DTT), another ER stress inducer (Supplementary Fig. 6). Similar to the observations based on TM treatment, *ire1a ire1b* showed a drastic reduction in shoot fresh weight while *bap2* exhibited a moderate but significant reduction in shoot fresh weight compared to Col-0 (Supplementary Fig. 6b). These results indicate that in conditions of UPR sufficiency BAP2 is required to enhance ER stress tolerance in cells challenged by chronic ER stress.

We next aimed to test whether *BAP2* could contribute to overcoming temporary ER stress. To this end, we performed an ER stress recovery assay[48] based on a TM-pulse (6 h) and drug-washout treatment, followed by root growth measurement after 4 days of growth on drug-free media (Supplementary Fig. 7). As a positive control, we used a *bzip28* knockout allele[49], which exhibits a strong reduction in root growth in recovery from temporary stress[31]. As expected, *bzip28* showed a significant inhibition in root growth compared to Col-0. In net contrast, the root growth of *bap2* was similar to Col-0, indicating that BAP2 is not necessary for recovery from ER stress and supporting the hypothesis that *BAP2* has a predominant role in unresolved ER stress conditions.

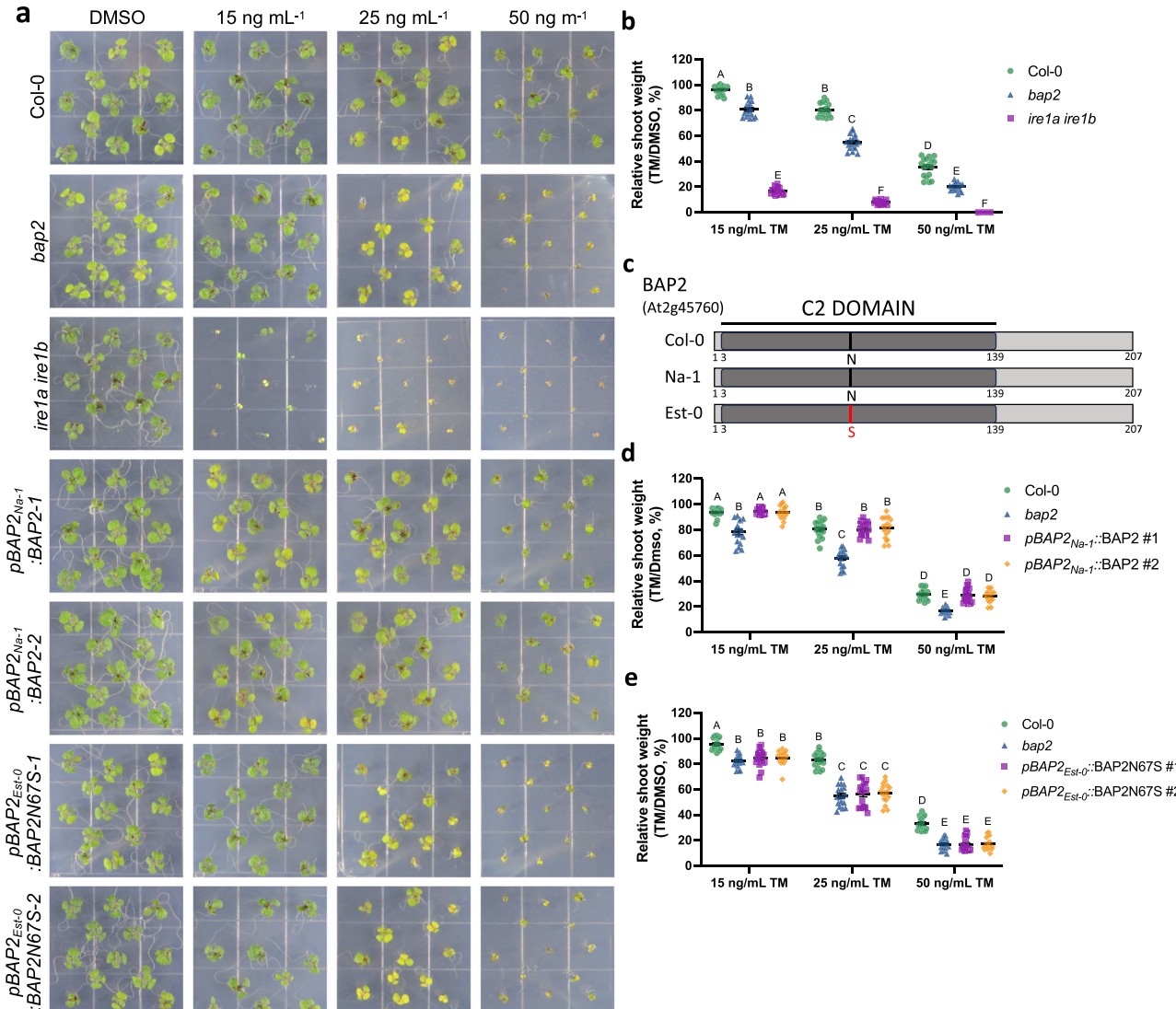

**Fig. 3 | BAP2 is necessary to overcome chronic ER stress, and the Na-1 allele, but not the Est-0 allele, complements *bap2* ER stress sensitivity. a** Representative images of Col-0, *bap2*, *ire1a ire1b* and the transgenic lines expressing Na-1 (*pBAP2_{Na-1}*:BAP2) or Est-0 (*pBAP2_{Est-0}*:BAP2N67S) genomic fragment (promoter and coding region) in the *bap2* background germinated on media containing the indicated concentrations of TM or DMSO grown for 10 days. **b** Relative shoot fresh weight of Col-0 (green dots), *bap2* (blue triangles), and *ire1a ire1b* (purple squares) seedlings treated as indicated in (**a**) Data represent means ± SEM among biological replicates (*n* = 18). Statistical significance was determined using a factorial linear mixed model framework followed by post-hoc testing using two-sided Tukey's HSD test (multiple testing-controlled thresholds used was *P* < 0.05).

**c** Diagram of Col-0, Na-1, and Est-0 BAP2 protein sequence. Mutation in the Est-0 sequence is indicated in red. The C2 domain is indicated in dark gray. **d**, **e** Relative shoot fresh weight of *pBAP2_{Na-1}*:BAP2 (**d**) or *pBAP2_{Est-0}*:BAP2N67S transgenic lines treated as indicated in (**a**) (Col-0 values are represented in green dots, *bap2* values in blue triangles, independent transgenic line #1 values in purple squares, and independent transgenic line #2 values in orange rhombuses). Data represent means ± SEM among biological replicates (*n* = 18). Statistical significance was determined using a factorial linear mixed model framework followed by post-hoc testing using two-sided Tukey's HSD test (multiple testing-controlled thresholds used was *P* < 0.05). Source data are provided as a Source Data file.

---

Since *BAP2* has partially overlapping functions with its homolog BAP1 in PCD[35], we next tested whether *BAP1* is also necessary to enhance the tolerance to chronic ER stress. To do so, we tested the sensitivity of a *bap1* knockout mutant[50] to TM-induced chronic ER stress and found that in contrast to *bap2*, *bap1* showed a similar ER stress response to Col-0 (Supplementary Fig. 8a–c). We also analyzed *BAP1* transcript levels at 6, 24, and 48 h of TM treatment, and found no significant differences compared to Col-0 (Supplementary Fig. 8d). Together, these results indicate that *BAP2* has assumed divergent functions from *BAP1*, at least in unresolved ER stress. Furthermore, because previous studies have shown BON1, a member of the evolutionarily conserved copine family[51], is a functional partner of BAP proteins[52], we analyzed the chronic ER stress sensitivity of a *bon1* knockout mutant[53,54]. *Bon1* showed a similar shoot fresh weight ratio to

Col-0 (Supplementary Fig. 9), consistent with previous results that the *bon1* mutant has a wild type-like response to chronic ER stress[55]. Together, these results indicate that *BAP2* has evolved a functional role in response to chronic ER stress, most likely through mechanisms that do not involve known functional interactors of BAP2.

## BAP2 Est-0 allele variation contributes to the ER stress sensitivity of Est-0

Having established that a *BAP2* knock-out mutation compromises positive ER stress outcomes in Col-0, we next sought to better understand how natural variation in the *BAP2* gene could contribute to the verified differences in ER stress response between Na-1 and Est-0 (Fig. 1b–d). Based on the 1001 Arabidopsis genome database[56], and as confirmed by Sanger sequencing, Na-1, and Est-0 have highly

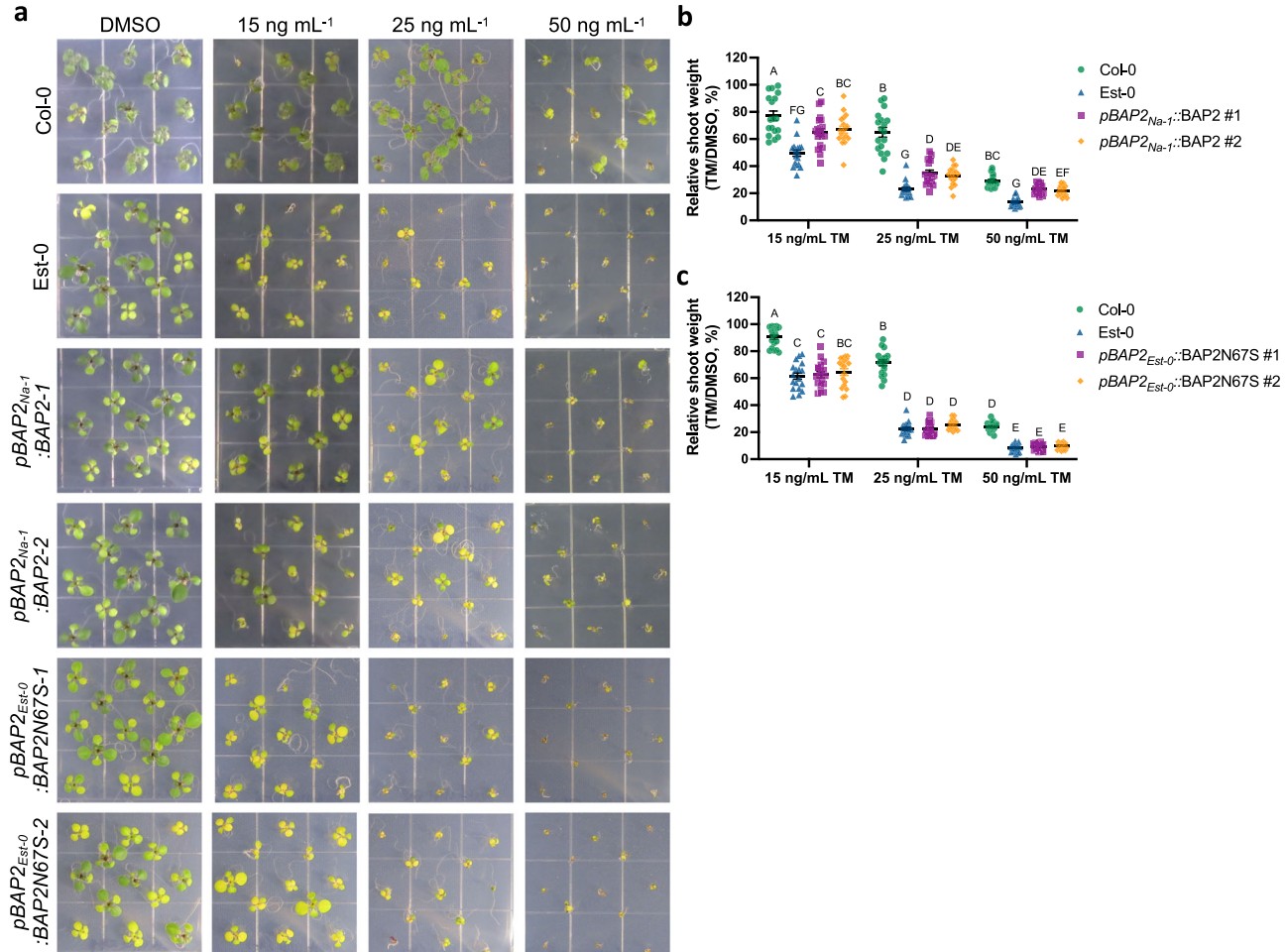

**Fig. 4 | The *BAP2* Est-0 allele contributes to Est-0 ER stress sensitivity while the Na−1 allele partially attenuates it. a** Representative images of Col-0, Est-0, and transgenic lines expressing Na-1 (*pBAP2_{Na-1}*:BAP2) or Est-0 (*pBAP2_{Est-0}*:BAP2N67S) genomic fragment (promoter and coding region) in Est-0 background seedlings germinated on media containing the indicated concentrations of TM or DMSO grown for 10 days. (**b**, **c**) Relative shoot fresh weight of *pBAP2_{Na-1}*:BAP2 (**b**) or *pBAP2_{Est-0}*:BAP2N67S (**c**) transgenic lines treated as indicated in (**a**) (Col-0 values are represented in green dots, *bap2* values in blue triangles, independent transgenic line #1 values in purple squares, and independent transgenic line #2 values in orange rhombuses). Data represent means ± SEM among biological replicates (*n* = 18) for relative shoot weight. Statistical significance was determined using a factorial linear mixed model framework followed by post-hoc testing using two-sided Tukey's HSD test (multiple testing-controlled thresholds used was *P* < 0.05). Source data are provided as a Source Data file.

conserved and similar sequences to Col-0 reference. However, in *BAP2*, we detected a single synonymous base pair change between the accessions and Col-0, and a non-synonymous change between Col-0 or Na-1 and Est-0. In Est-0, the non-synonymous base pair change causes a substitution of Asp to Ser at position 67 compared to Col-0 and Na-1 sequences (Fig. 3c and Supplementary Fig. 10a). This substitution occurs in the C2 functional domain of BAP2, which is highly conserved with its homolog, *BAP1* (Fig. 3c and Supplementary Fig. 10b)[35,52]. To test whether these *BAP2* mutations contribute to the different sensitivity of the accessions to ER stress, we transformed *bap2* (Col-0 background) with either the Na-1 or Est-0 *BAP2* genomic clone, which included the 1 kb of promoter region from the start codon of the respective genes (*pBAP2_{Na-1}*:BAP2, and *pBAP2_{Est}*:BAP2N67S, respectively, Supplementary Fig. 11). We then analyzed the response to ER stress in the complemented lines by measuring shoot fresh weight (Fig. 3a, d, e and Supplementary Fig. 3b, c) and chlorophyll content (Supplementary Figs. 4b, 5b, 5c). Analyses of two independent lines expressing *pBAP2_{Na-1}*:BAP2 indicated complementation of the ER stress sensitivity of the *bap2* mutant (Fig. 3d), as expected based on the identity of the *BAP2* sequences in Col-0 and Na-1. Furthermore, the analyses confirmed that the loss-of-function of *BAP2* is responsible for the increased

sensitivity of *bap2* mutant to TM. In contrast, two independent lines expressing *pBAP2_{Est}*:BAP2N67S, displayed an ER stress sensitivity similar to the *bap2* mutant (Fig. 3e). Together these results demonstrate a reduced function of the Est-0 *BAP2* allele in ER stress and support the hypothesis that *BAP2* is a main factor for the ER stress sensitivity of the Chr 2 QTL.

To further explore the involvement of *BAP2* in the ER stress sensitivity of Est-0, we expressed *pBAP2_{Na-1}*:BAP2, and *pBAP2_{Est}*:BAP2N67S in the Est-0 background and tested the sensitivity to chronic ER stress (Fig. 4). While the Est-0 transgenic lines expressing *pBAP2_{Est}*:BAP2N67S had a response similar to Est-0 (Fig. 4c and Supplementary Figs. 3e, 4e, 5e), the Est-0 lines transformed with *pBAP2_{Na-1}*:BAP2 showed a significantly enhanced ER stress resistance compared to the Est-0 background and the Est-0 lines expressing *pBAP2_{Est}*:BAP2N67S (Fig. 4b and Supplementary Figs. 3d, 4d, 5d). All the analyzed transgenic lines showed similar *BAP2* expression levels compared to either Col-0 or Est-0 (Supplementary Fig. 12a, b), indicating that differences in ER stress response are due to *BAP2* sequence variation rather than differences in expression of the transgenes. We also tested whether Na-1 or Est-0 had different expression levels of *BAP2* under ER stress. We found no substantial differences between them or compared to Col-0

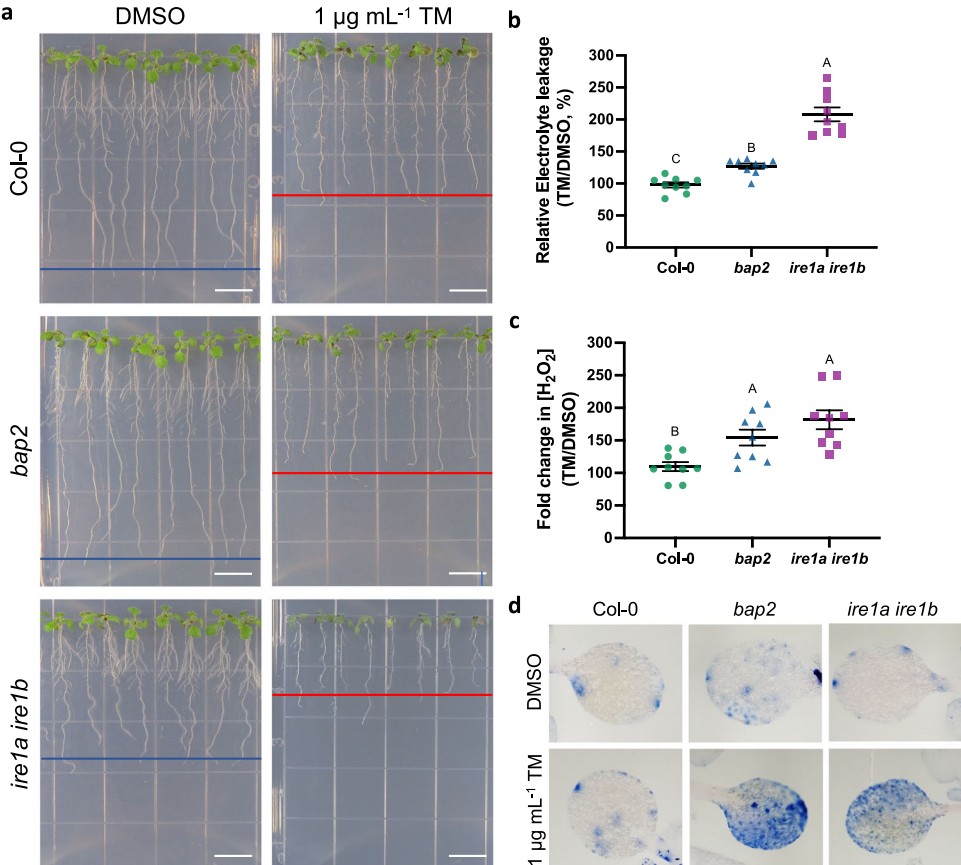

**Fig. 5 | *BAP2* functions as a negative regulator of ER stress-induced cell death.**
**a** Representative images of 7-day-old Col-0, *bap2*, and *ire1a ire1b* seedlings treated with 1 µg mL⁻¹ TM or DMSO for 48 h that were used to determine the extent of cell death by quantification of the relative percent electrolyte leakage (**b**) or the relative percent of accumulation of $H_2O_2$ quantified by an Amplex Ultra Red assay (**c**). Col-0 values are represented in green dots, *bap2* values in blue triangles, *ire1a ire1b* values in purple squares. Data represent mean ± SEM among biological replicates (*n* = 9).

Statistical significance was determined using a factorial linear mixed model framework followed by post-hoc testing using two-sided Tukey's HSD test (multiple testing-controlled thresholds used was $P < 0.05$). (Scale bar = 1 cm).
**d** Representative cotyledons of 7-day-old Col-0, *bap2*, and *ire1a ire1b* seedlings treated as indicated in (**a**) and stained with trypan blue dye. Scale bar = 1 cm. Source data are provided as a Source Data file.

(Supplementary Fig. 12c). Together these results support our hypothesis that *BAP2* variation is involved in the ER stress sensitivity of Est-0. Interestingly, the $pBAP2_{Na-1}$:BAP2 complemented Est-0 lines showed only partial complementation compared to Col-0 (supplementary Fig. 3c), suggesting that Est-0 increased sensitivity to TM may be contributed by additional factors, such as other gene variations at the same or another QTL. Nonetheless, the complementation of *bap2* sensitivity to ER stress by Na-1 *BAP2* coding sequence, which is identical to Col-0 *BAP2* (Supplementary Fig. 11), supports our results that *BAP2* is required to enhance the tolerance to chronic ER stress (Fig. 3a, d and Supplementary Figs. 3b, 4b, 5b).

**In conditions of UPR sufficiency, BAP2 is a negative regulator of PCD during chronic ER stress**
*BAP2* functions as a regulator of PCD caused by several abiotic and biotic conditions[35]. Therefore, we hypothesized that BAP2 could also be involved in regulating cell death caused by ER stress. To test this, we measured the extent of cell death by quantification of electrolyte leakage[57] in Col-0, *bap2*, and *ire1a ire1b* 7-days-old seedlings treated with 1 mg mL⁻¹ TM or DMSO for 48 h (Fig. 5a, b). Compared to WT, *ire1a ire1b* (positive control) and *bap2* showed a significant increase in electrolyte leakage (Fig. 5b and Supplementary Fig. 13a)[58]. To further examine the ER stress-induced cell death of the *bap2* mutant, we stained 7-days-old Col-0, *bap2*, and *ire1a ire1b* seedlings treated with 1 mg mL⁻¹ TM or DMSO with trypan blue, which only stains dead cells[59]. Supporting the electrolyte leakage results, we found the TM-treated

shoots of *bap2* and *ire1a ire1b* seedlings showed more extended trypan blue-positive areas compared to Col-0 (Fig. 5d and Supplementary Fig. 14). Therefore, BAP2 is required to antagonize PCD in conditions of UPR sufficiency in ER stress.

It has been proposed that $H_2O_2$-induced cell death is negatively regulated by *BAP2*[35], and that ER stress causes accumulation of $H_2O_2$ in Arabidopsis[60,61]. Therefore, we next examined the $H_2O_2$ levels in 7-days-old Col-0, *bap2*, and *ire1a ire1b* (control) seedlings treated with 1 mg mL⁻¹ TM or DMSO for 48 hours (Fig. 5c and Supplementary Fig. 13b). By using the Amplex Ultra Red (AUR) assay, a sensitive enzyme-based fluorimetric assay[58,62], we found a significant increase in $H_2O_2$ levels in *ire1a ire1b*, as previously reported[58]. Interestingly, *bap2* also showed significant increase in $H_2O_2$ levels compared to Col-0, but no differences with *ire1a ire1b* (Fig. 5c and Supplementary Fig. 13b), indicating that, similarly to IRE1, BAP2 functions to attenuate $H_2O_2$ accumulation during ER stress.

Because our data indicate that the Est-0 genome encodes a *BAP2* allele that is unable to complement the ER stress sensitivity of *bap2* (Figs. 3e, 4c), we next tested whether this was also due to the loss in the ability to regulate cell death and $H_2O_2$ accumulation in ER stress conditions. To do so, we measured electrolyte leakage and $H_2O_2$ accumulation in the *bap2* transformed lines expressing either $pBAP2_{Na-1}$:BAP2, or $pBAP2_{Est}$:BAP2N67S treated with 1 mg mL⁻¹ TM or DMSO for 48 h (Supplementary Fig. 15). We did not find significant differences in the $pBAP2_{Na-1}$:BAP2 transgenic lines compared to Col-0, as expected based on the complementation of the ER stress sensitivity (Fig. 3a, d).

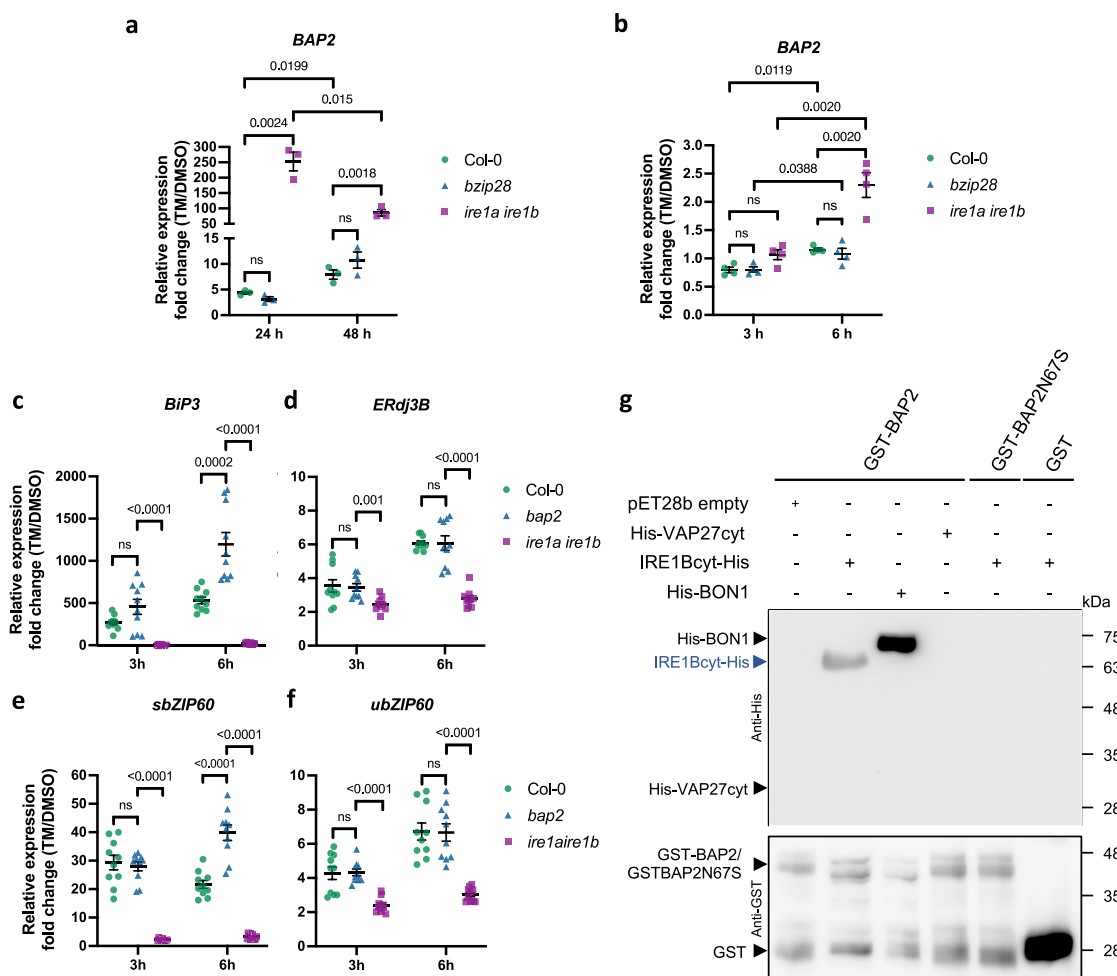

**Fig. 6 | *BAP2* induction is controlled by IRE1, and it negatively modulates UPR activation under ER stress. a** qRT-PCR analyses of *BAP2* expression in Col-0 (green dots), *bzip28* (blue triangles), and *ire1a ire1b* (purple squares) 5-day-old seedlings growth during 24 and 48 h in plates containing 1 μg mL⁻¹ TM or DMSO. Values are presented relative to the indicated DMSO control. Transcription of *UBQ10* was used as an internal control. Data represent mean ± SEM among biological replicates (*n* = 3). Statistical significance was determined by Student's unpaired two-tailed *t* test (*P*-value is shown in figure; ns, not significant). **b** qRT-PCR analyses of *BAP2* expression in Col-0 (green dots), *bzip28* (blue triangles), and *ire1a ire1b* (purple squares) seedlings after 3- or 6- h pulse treatment with 0.5 μg mL⁻¹ TM or DMSO. Values are presented relative to the indicated DMSO control. Transcription of *UBQ10* was used as an internal control. Data represent mean ± SEM among biological replicates (*n* = 4). Statistical significance was determined by Student's

unpaired two-tailed *t*-test (*P*-value is shown; ns, not significant). **c–f** qRT-PCR analyses of UPR target genes *BiP3* (**c**), *ERdj3B* (**d**), *sbZIP60* (**e**), and *ubZIP60* (**f**) in Col-0 (green dots), *bap2* (blue triangles), and *ire1a ire1b* (purple squares) seedlings after 3- h or 6- h pulse treatment with 0.5 μg mL⁻¹ TM or DMSO. Values are presented relative to the indicated DMSO control. Transcription of *UBQ10* was used as an internal control. Data represent mean ± SEM among biological replicates (*n* = 10). Statistical significance was determined by Student's unpaired two-tailed *t* test (*P*-value is shown; ns, not significant). **g** In vitro interaction assay between GST-BAP2 or GST-BAP2N67S and IRE1Bcyt-His. The pull-down experiment using anti-GST resin was analyzed by Western blot. BON1-His was used as positive control while His-VAP27cyt and empty pET28b were used as negative controls. The interaction between GST alone and IRE1Bcyt-His was also analyzed. Source data are provided as a Source Data file.

In net contrast, *bap2* transgenic lines expressing *pBAP2ₑₛₜ*:BAP2N67S showed similar levels of electrolyte leakage and H₂O₂ accumulation compared to *bap2*. These results are consistent with the hypothesis that *BAP2* variation is significantly responsible for Est-0 ER stress sensitivity (Fig. 4c) and support our results that an intact C2 domain of BAP2 is necessary for the ER stress-responsive pro-survival functions of BAP2 in conditions of UPR sufficiency (Fig. 3e).

**The levels of *BAP2* expression in ER stress are controlled by IRE1 but not bZIP28**

Our results thus far indicate that *BAP2* is induced in prolonged ER stress in Col-0 (Fig. 2d). Therefore, we next aimed to define the UPR signaling arm that controls such an induction. To do so, we compared *BAP2* mRNA levels at 24 or 48 h of TM treatment in *ire1a ire1b* and *bzip28* mutants with Col-0 by qRT-PCR (Fig. 6a). As control of UPR activation, we examined *BiP3* expression levels

(Supplementary Fig. 16a). We found a similar increase of *BAP2* expression over time in Col-0 and the *bzip28* mutant, supporting that *BAP2* induction is independent of bZIP28. Surprisingly, we also observed a sharp increase in *BAP2* transcript levels in the *ire1a ire1b* mutant at 24 and 48 h of TM treatment (Fig. 6a), suggesting that IRE1 functions to suppress *BAP2* induction in ER stress. Based on these results, we next aimed to test whether IRE1 could be involved in controlling the levels of *BAP2* during the adaptive phase of ER stress, and monitored *BAP2* induction levels in Col-0, *ire1a ire1b,* and *bzip28* at 3 or 6 hours of TM treatment (Fig. 6b). As a control of UPR activation, we examined *BiP3* expression levels (Supplementary Fig. 16b). We did not find a significant *BAP2* induction in Col-0 and *bzip28*; conversely, we found a small but significant increase in *BAP2* transcript levels in *ire1a ire1b* at 6 h of TM treatment and no differences at 3 h of treatment. Together, these results indicate that *BAP2* induction during ER stress is independent of bZIP28 and that

the loss of IRE1 leads to an increase in *BAP2* transcription as ER stress progresses.

## BAP2 attenuates the induction of UPR genes in adaptive ER stress through the IRE1 signaling pathway

Because our results indicate that *bap2* has an increased sensitivity to ER stress (Fig. 3a, b) and that IRE1 suppresses the expression of *BAP2* in ER stress (Fig. 6a, b), we hypothesized that the loss of *BAP2* could affect UPR gene expression under ER stress. To do so, we analyzed the transcript levels of *BiP3*, *ERdj3B*, and *sbZIP60* in Col-0, *ire1a ire1b*, and *bap2* upon 3 h or 6 h TM treatment by qRT-PCR (Fig. 6c–f). Consistent with previous reports[18,31,58], the *ire1a ire1b* mutant showed a drastic reduction in *BiP3* and *sbZIP60* transcript levels compared to Col-0 and a slight but significant reduction in *ERdj3B* expression levels after 6 h of TM treatment (Fig. 6c–e). In contrast, we found that the induction of *BiP3* and *sbZIP60* was significantly higher in *bap2* relative to Col-0 upon 6 h of TM treatment (Fig. 6c, e). No significant changes were observed in *ERdj3B* levels (Fig. 6d). To investigate if the increase of *sbZIP60* transcripts was due to an increase in *bZIP60* expression, we also analyzed the transcript levels of the unspliced form of *bZIP60* (*ubZIP60*) (Fig. 6f). We found similar *ubZIP60* levels in *bap2* and Col-0, indicating that the verified *sbZIP60* increased levels in *bap2* are likely due to an increase in IRE1 splicing activity. Together with the notion that in ER stress conditions, *BiP3* expression is controlled preferentially by IRE1-bZIP60[31] and the evidence that BAP2 negatively regulates *BiP3* induction (Fig. 6c), these results support that during ER stress BAP2 negatively modulates the activation of UPR genes that are primarily regulated by the IRE1-bZIP60 arm.

Our results suggest that BAP2 negatively modulates IRE1 splicing activity (Fig. 6e, f). Therefore, we next aimed to analyze if BAP2 regulates IRE1 function at a transcriptional level by analyzing the transcript levels of *IRE1A* and *IRE1B* in Col-0 and *bap2* upon 6 h TM treatment by qRT-PCR (Supplementary Fig. 17). We did not observe a significant difference in *IRE1A* or *IRE1B* transcript levels in *bap2* compared to Col-0, suggesting that BAP2 does not modulate IRE1 activity at a transcriptional level. Then, to test if BAP2 could regulate IRE1 directly, we conducted a test for an interaction between BAP2 and IRE1 in vitro (Fig. 6g). To do so, we performed an interaction assay using the predominant IRE1 paralog, IRE1B[18]. Because IRE1 is a type I membrane-associated protein, we used the cytosolic region of IRE1, which contains the kinase and endonuclease domains, fused to a His6x-Tag at the C-terminal (IRE1Bcyt-His), and BAP2 or BAP2N67S fused to GST at the N-terminus (GST-BAP2). As controls for BAP2, we combined GST-BAP2 with the known BAP2 interactor BON1 (His-BON1; positive control), the cytosolic domain of the ER-membrane associated VAP27-1 fused to His (His-VAP27cyt; negative control)[63], and p28b empty vector (for His-tag production; negative control). For IRE1, we combined IRE1Bcyt-His with GST (Fig. 6g and Supplementary Fig. 18). As expected, GST-BAP2 and His-BON1 interacted with each other, but BAP2-GST did not interact with His-VAP27cyt (Fig. 6g). For the GST-BAP2 and IRE1Bcyt-His combination, we found an in vitro association between both proteins (Fig. 6g). However, we did not observe an interaction between IRE1Bcyt-His and GST-BAP2N67S (Fig. 6g). These results suggest that BAP2 may interact with IRE1B to modulate its activity, and that the structure of BAP2 C2 domain may be involved in this interaction.

## BAP2 is a pro-death effector when the UPR is insufficient to respond to ER stress

So far, our results show that in conditions of UPR sufficiency in chronic ER stress, BAP2 acts as a negative modulator of the expression of UPR genes primarily controlled by the IRE1-bZIP60 arm (Fig. 6c–f) and that the IRE1 functions to suppress *BAP2* induction (Fig. 6a, b). These results support an antagonistic yet balancing, relationship between IRE1 and BAP2 to enhance resistance to ER stress. Therefore, we next aimed to

test the functional role of BAP2 in chronic ER stress during UPR insufficiency, which is mimicked by the loss of IRE1 and leads to the execution of PCD[11,18]. To do so, we generated an *ire1a ire1b bap2* triple mutant. Then, we grew Col-0, *bap2*, *ire1a ire1b*, and *ire1a ire1b bap2* seedlings for 10 days on media containing 25 or 50 ng mL⁻¹ of TM for chronic ER stress induction and estimated the ratio of shoot fresh weight compared to mock treatment. We found that *ire1a ire1b* is lethal in chronic ER stress (Supplementary Fig. 19), as expected[11,18,19]. However, we also found a slight but significant increase of ER stress tolerance in *ire1a ire1b bap2* mutant at 25 ng mL⁻¹ of TM concentration, but not at 50 ng mL¹ of TM concentration, suggesting a pro-death function of BAP2 in ER stress conditions when IRE1 signaling is insufficient or dysfunctional. To test this hypothesis, we grew Col-0, *bap2*, *ire1a ire1b*, and *ire1a ire1b bap2* seedlings for 10 days on media containing lower concentrations of TM (i.e., 5, 10 or 20 ng mL⁻¹) to induce prolonged but mild ER stress and measured the ratio of shoot fresh weight and chlorophyll content compared to mock treatment (Fig. 7a, b and Supplementary Fig. 20). We found that in these conditions, *ire1a ire1b bap2* partially reverted the lethal phenotype of *ire1a ire1b* with a significantly marked increase in shoot fresh weight (Fig. 7b) and chlorophyll content (Supplementary Fig. 20b, c) compared to *ire1a ire1b*. Because we have previously found that BAP2 suppresses PCD under chronic ER stress when IRE1 is functional (Fig. 5b), we postulated that regulation of PCD could be involved in the observed *ire1a ire1b bap2* phenotype. To this end, we measured electrolyte leakage in Col-0, *bap2*, *ire1a ire1b*, and *ire1a ire1b bap2* seedlings grown on media containing 5, 10 or 20 ng mL⁻¹ of TM for 10 days (Fig. 7c). We observed a reduced increase of the electrolyte leakage in *ire1a ire1b bap2* mutant compared to *ire1a ire1b*. These data indicate a pro-death role for BAP2 in prolonged ER stress in conditions of UPR insufficiency.

## When IRE1 is dysfunctional, BAP2 regulates H2O2 accumulation under prolonged ER stress conditions

Our results indicate that BAP2 is involved in maintaining low levels of $H_2O_2$ accumulation when IRE1 is functional under ER stress conditions (Fig. 5c). Therefore, we next tested if the different ER stress sensitivity phenotype observed in *ire1a ire1b* and *ire1a ire1b bap2* mutants was related to a different $H_2O_2$ accumulation. To do so, we measured the $H_2O_2$ levels in Col-0, *bap2*, *ire1a ire1b*, and *ire1a ire1b bap2* seedlings grown on media containing 5, 10, or 20 ng mL⁻¹ of TM for 10 days by AUR assay. We did not find significant differences in $H_2O_2$ levels between *ire1a ire1b* and *ire1a ire1b bap2* (Supplementary Fig. 21). However, we might not have seen small differences in these conditions because of the extensive cell damage caused by prolonged ER stress. Therefore, we conducted an AUR assay in 7-days-old Col-0, *bap2*, and *ire1a ire1b* seedlings treated with 0.25 mg mL⁻¹, 0.5 mg mL⁻¹ or 1 mg mL⁻¹ TM or DMSO for 48 h (Fig. 7d and Supplementary Fig. 22a). Under these conditions, we found decreased $H_2O_2$ levels in *ire1a ire1b bap2* compared to *ire1a ire1b* at 0.25 mg mL⁻¹, 0.5 mg mL⁻¹ TM concentrations, but not at 1 mg mL⁻¹ TM concentration (Fig. 7d). Then we measured electrolyte leakage levels (Fig. 7e) and relative fresh weight (Supplementary Fig. 22b) in Col-0, *bap2*, *ire1a ire1b*, and *ire1a ire1b bap2* seedlings. Similar to the $H_2O_2$ accumulation levels, we found a decrease of electrolyte leakage at 0.25 mg mL⁻¹ and 0.5 mg mL⁻¹ TM concentrations, but not at 1 mg mL⁻¹ TM concentration (Fig. 7e). Therefore, our results indicate that BAP2 induces $H_2O_2$ levels during prolonged ER stress when IRE1 is not functional. Together with the evidence that under these conditions the levels of $H_2O_2$ increased in the *bap2* and *ire1a ire1b bap2* mutants (Fig. 7e), these results indicate that BAP2 is a key regulator of $H_2O_2$ accumulation and PCD under ER stress conditions. Specifically, BAP2 functions as a suppressor of PCD and $H_2O_2$ production when IRE is sufficient but acts as a pro-death effector leading to $H_2O_2$ accumulation and PCD when IRE1 signaling is insufficient.

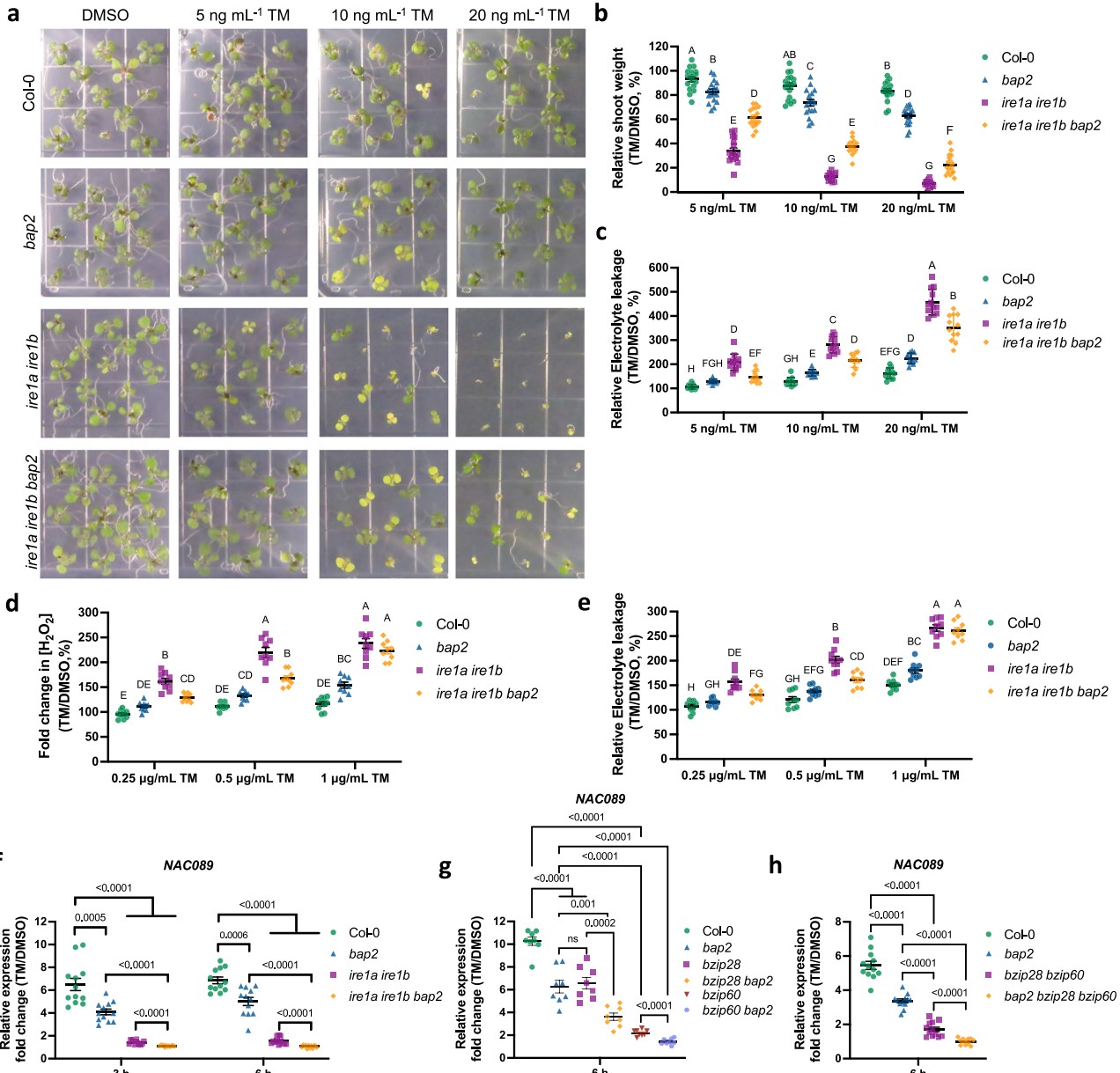

**Fig. 7 | BAP2 is a pro-death effector when IRE1 is dysfunctional. a** Representative images of Col-0, *bap2, ire1a ire1b,* and *ire1a ire1b bap2* seedlings germinated on media containing the indicated concentrations of TM or DMSO for 10 days. **b** Relative shoot fresh weight of seedlings treated as indicated in (**a**). **c** Quantification of the relative percent electrolyte leakage to determine the extent of cell death in Col-0, *bap2, ire1a ire1b,* and *ire1a ire1b bap2* seedlings treated as indicated in (**a**). **d, e** Quantification of relative percent of accumulation of $H_2O_2$ by an Amplex Ultra Red assay (**d**) and the relative percent electrolyte leakage (**e**) in Col-0, *bap2, ire1a ire1b,* and *ire1a ire1b bap2* seedlings treated with 1 µg mL⁻¹ TM or DMSO for 48 hours. **b–e** Data represent mean ± SEM among biological replicates ($n = 28$ for (**b**), $n = 12$ for (**c**), $n = 9$ for (**d**) and (**e**). Statistical significance was determined using a factorial linear mixed model framework followed by post-hoc testing using two-sided Tukey's HSD test (multiple testing-controlled threshold used was $P < 0.05$). Color legend: Col-0 in green dots, *bap2* in blue triangles, *ire1a* *ire1b* in purple squares, and *ire1a ire1b bap2* in orange rhombuses. **f, g** qRT-PCR analysis of *NAC089* in Col-0 (green dots), *bap2* (blue triangles), *ire1a ire1b* (purple squares), and *ire1a ire1b bap2* (orange rhombuses) seedlings after 3- h or 6- h pulse treatment with 0.5 µg mL⁻¹ TM or DMSO (**f**) or in Col-0 (green dots), *bap2* (blue triangles), *bzip28* (purple squares), *bzip60* (orange rhombuses), *bzip28 bap2* (brown invert triangles) and *bzip60 bap2* (violet hexagons) seedlings after 6 h pulse treatment with 0.5 µg mL⁻¹ TM or DMSO (**g**) or Col-0 (green dots), *bap2* (blue triangles), *bzip28 bzip60* (purple squares) and *bzip28 bzip60 bap2* (orange rhombuses) seedlings after 6 h pulse treatment with 0.5 µg mL⁻¹ TM or DMSO (**h**). Values are presented relative to the indicated DMSO control. Transcription of *UBQ10* was used as an internal control. Data represent mean ± SEM among biological replicates ($n = 12$ for (**f**), $n = 8$ for (**g**) and (**h**). Statistical significance was determined by Student's unpaired two-tailed *t*-test (*P*-value is shown; ns, not significant). Source data are provided as a Source Data file.

## BAP2 induces the expression of the PCD-inducer NAC089 under ER stress

Since in ER stress, NAC089 has been identified as a critical promotor of PCD by inducing the transcription of genes involved in PCD[34], we next examined the expression of *NAC089* in Col-0, *ire1a ire1b, bap2,* and *ire1a ire1b bap2* upon 3 h or 6 h TM treatment by qRT-PCR. We found

that *NAC089* transcripts were induced in ER stress in Col-0 but that the loss of either BAP2 or IRE1 led to a decreased induction of *NAC089* transcription compared to Col-0 (Fig. 7f). These results indicate that BAP2 and IRE1 are necessary for a homeostatic induction of *NAC089* in ER stress conditions. We also found that *NAC089* transcription levels were abolished in the *ire1a ire1b bap2* mutant, suggesting that IRE1 and

BAP2 induce the expression of *NAC089* in ER stress through largely independent mechanisms. Because the transcriptional regulation of *NAC089* is mediated by bZIP60[34], we next aimed to investigate the requirement of bZIP60 in the BAP2-mediated induction of *NAC089*. To do so, we generated a *bzip60 bap2* double mutant. Then, we analyzed the transcription levels of *NAC089* in Col-0, *bap2, bzip60,* and *bzip60 bap2* after 6 h TM treatment by qRT-PCR (Fig. 7g). Similar to the results observed in *ire1a ire1b* and *ire1a ire1b bap2* mutants (Fig. 7f), *bzip60* showed a strongly reduced *NAC089* transcription levels while *NAC089* induction was abolished in *bzip60 bap2*, supporting the hypothesis that BAP2 and IRE1-bZIP60 have partially overlapping functions in the regulation of *NAC089* induction consistent with the verified *NAC089* induction in *ire1a ire1b bap2*.

*NAC089* activation under ER stress also involves bZIP28 which, like bZIP60, binds *NAC089* promoter to directly control expression[34]. To test if bZIP28 is involved in the induction *NAC089* controlled by BAP2 under ER stress conditions, we generated a *bzip28 bap2* mutant, and we investigated *NAC089* transcription levels in Col-0, *bap2, bzip28,* and *bzip28 bap2* after 6 h TM treatment by qRT-PCR (Fig. 7g). Our results showed a similar decrease in *NAC089* transcription levels in *bap2* and *bzip28* compared to Col-0 while the *bzip28 bap2* double mutant showed a higher decrease in *NAC089* induction under ER stress, indicating that BAP2 and bZIP28 control *NAC089* induction under ER stress through different mechanisms. Finally, to confirm our results, we generated a *bzip28 bzip60 bap2* triple mutant. Then, we investigated the levels of *NAC089* induction after 6 h TM treatment in Col-0, *bap2, bzip28 bzip60,* and *bzip28 bzip60 bap2* by qRT-PCR (Fig. 7h). Consistent with our initial results, *bzip28 bzip60* showed a greater inhibition of *NAC089* induction under ER stress, similar to the inhibition observed previously in *ire1a ire1b* and *bzip60*, while *bzip28 bzip60 bap2* showed the strongest reduction in *NAC089* transcription levels, similar to the levels observed in *ire1a ire1b bap2* and *bzip60 bap2*, further supporting the hypothesis that BAP2 and IRE1 control *NAC089* induction in ER stress through only partially overlapping mechanisms and that BAP2 is sufficient to induce *NAC089* in conditions of IRE1 dysfunction.

## Discussion

Plant natural variation is a valuable resource to gain insights into complex pathways and pinpoint effect-generating polymorphisms in responses to a variety of stress cues[36]. In this work, *A. thaliana* natural genetic variation allowed us to identify the PCD regulator BAP2 as a critical plant UPR modulator and to implicate the *BAP2* SNP polymorphism with the contrasting ER stress sensitivity in two ecotypes, Est-0 and Na-1. *BAP2* is induced by abiotic and biotic stress[35,52]. Our work indicates that *BAP2* is also induced by ER stress in an IRE1-regulated manner and that BAP2 is necessary to control the activity of the UPR in ER stress responses. Therefore, our results support a broader role of BAP2 in stress responses than previously thought and identify BAP2 as the long sought-after rheostat between PCD and the plant UPR that is required to modulate the activity of the UPR master regulator IRE1 in ER stress and execute pro-death processes in conditions of UPR insufficiency (Fig. 8).

Studying natural variation provides an avenue for exploring mechanisms of complex pathways, such as the UPR. Indeed, a previous analysis of *Drosophila* genetic variation identified genes putatively involved in ER stress response[64]. In this work, natural genetic variation in *A. thaliana* allowed the discovery of BAP2 as a critical UPR regulator.

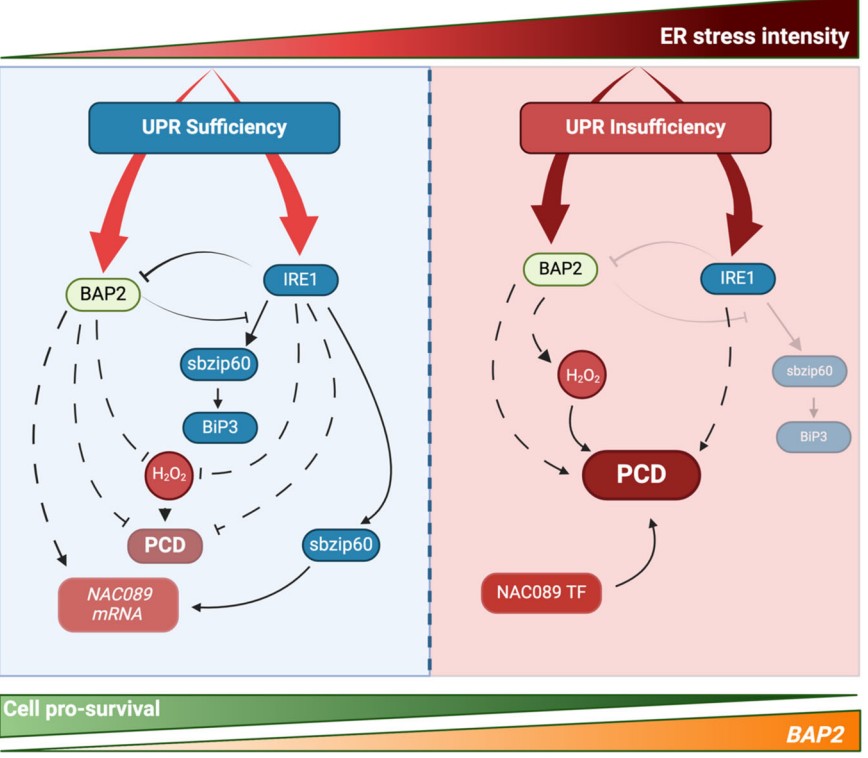

**Fig. 8 | Schematic model illustrating the functional interaction between BAP2 and IRE1 in cell fate determination during ER stress.** During the adaptive phase of the UPR, IRE1 activates pro-survival strategies to outweigh constitutive pro-death processes. In this model, BAP2 functions as a rheostat that monitors IRE1 function to prevent irreversible outcomes, while IRE1 controls *BAP2* levels as a negative feedback loop for an optimization of UPR sufficiency. During this phase, IRE1 and BAP2 may induce *NAC089* expression as a preventive process for NAC089 requirement to ignite PCD when the UPR becomes insufficient in unresolved ER stress. When the latter takes place, BAP2 acts as a pro-death effector, leading to $H_2O_2$ accumulation and PCD as an irreversible step that outweighs pro-life adaptations. Therefore, cell-fate determination under ER stress is the result of a tug-of-war between pro-life and pro-death processes. Continuous arrows indicate direct known effects and dotted arrows indicate indirect effects. Created with BioRender.com.

Moreover, we showed that a single non-synonymous change in the C2 functional domain of BAP2 significantly contributes to contrasting ER stress sensitivity in two ecotypes, Est-0 and Na-1. However, the partial complementation observed in the $pBAP2_{Na-1}$:BAP2 complemented Est-0 line (Fig. 4b) supports that other factors are involved in the Est-0 increased sensitivity to TM and is consistent with our results that Est-0 is more sensitive to ER stress (RR = 0.21 ± 0.03; Fig. 1d) than the $bap2$ mutant (RR = 0.62 ± 0.05; Fig. 2b). In addition, we selected $BAP2$ as a candidate based on its transcriptional response under ER stress conditions. Therefore, it is possible that other genes present in our QTL or in other genome locations, which could be regulated at a different level, could also contribute to the observed variations of ER stress sensitivity among the different ecotypes.

The adaptive phase of the UPR is defined by the activation of pro-life responses; when these become insufficient or dysregulated, irreversible pro-death processes outweigh the pro-life adaptations and result in cell death[2,21]. In this model, ER stress survival is a tug-of-war between pro-life and pro-death processes orchestrated by UPR regulators, such as IRE1, and their tight regulation. The mechanisms that control IRE1 activity in plants are largely unknown. In conditions of UPR homeostasis, plant IRE1 is sufficient for ER stress survival. However, the loss of two $Arabidopsis$ IRE1 paralogs leads to UPR insufficiency and accelerates cell death under chronic ER stress, indicating that IRE1 is essential to the activation of pro-survival strategies to outweigh pro-death processes (Fig. 8). The latter take over as cell-fate determinants when IRE1 is nonfunctional and the UPR becomes insufficient, as in the $ire1a$ $ire1b$ mutant[11,18,19]. Such a critical role of a UPR regulator in cell fate determination requires the amplitude and duration of the UPR to be tightly regulated by molecular rheostats that monitor variations in ER stress intensity and adjust response levels to prevent irreversible outcomes. In mammalian cells, the three ER stress sensors are regulated through post-translational modifications and the binding of regulators[21], such as BI-1, which attenuate IRE1a signaling in chronic ER stress[30]. However, $Arabidopsis$ BI-1 does not reduce IRE1 splicing activity[31]. Other IRE1 modulators, such as SPL6[65], BL1[66], and GAAP1/GAAP3[67,68] have been identified. However, SPLI and BLI have been proposed to function by controlling IRE1 functions under normal growth conditions, in the case of SPL1 via induction of IRE1 expression[65,66]. GAPP1/GAAP3 have been reported to interact directly with IRE1[67] and cause an anti-apoptotic effect[68]; however, their function in cell-fate determination has not been established yet.

In mammalian cells, cell fate under prolonged ER stress depends on the coordination between the PERK and the IRE1 pathways[69,70]. In plants, IRE1 has been proposed to be crucial in cell-fate determination under ER stress conditions; however, the mechanism by which IRE1 determines cell fate under chronic mild conditions remains poorly understood. Our study provides evidence that during ER stress adaptation, BAP2 does not modulate bZIP28 activity but, rather, attenuates $bZIP60$ splicing, which is mediated by IRE1[31] (Fig. 8). We also found an interaction between BAP2 and IRE1B in vitro. We speculate that BAP2 might regulate IRE1 activity by a direct binding, similar to the binding of metazoan BI and IRE1, which reduces IRE1 activity[30]. Furthermore, we found that BAP2 is necessary to monitor the adaptive UPR by controlling the amplitude of ER stress gene expression primarily regulated by the IRE1-bZIP60 arm. These results and the genetic evidence provided in this work suggest that IRE1 operates dependently on BAP2 for UPR gene induction and chronic ER stress sensitivity indicates that BAP2 represses IRE1 by functioning as an IRE1 activity regulator (Fig. 8), analogously to BI-1 in metazoan cells[30]. Under this light, we propose that the loss of BAP2 is conducive to enhanced hypersensitivity to chronic ER stress because of IRE1 misregulation, which compromises the homeostatic balance between pro-life and constitutive pro-death pathways, in favor of the latter. Our results also show that in ER stress $BAP2$ is induced in a manner that is antagonized by IRE1 (Fig. 6a, b). These results and the evidence that the expression of UPR

genes controlled by IRE1-bZIP60 (i.e., $BiP3$[31]) is enhanced in $bap2$ suggest that $BAP2$ levels may be controlled by IRE1 in a negative feedback loop to enhance IRE1 activity to respond to ER stress through a reduction of the levels of an IRE1-function attenuator (i.e., BAP2). In this model (Fig. 8), in conditions of UPR sufficiency in ER stress, BAP2 would function as a rheostat to monitor the IRE1 function, which, in turn, would control $BAP2$ levels for an optimization of UPR sufficiency.

The documented suppression of the $H_2O_2$-induced cell death in yeast by the Arabidopsis BAP2 and the accumulation of $H_2O_2$ in the $bap1$ and $bap2$ mutant combinations[35] indicate that BAP proteins are required to control ROS homeostasis during stress responses. Consistently with these findings, our results indicate that $bap2$ has increased levels of $H_2O_2$ under chronic ER stress compared to wild type. In ER stress, ROS accumulation increases, leading to oxidative stress and PCD. However, low concentrations of ROS are required for signaling processes[61,71]. In plants, very little is known about the mechanisms for the dual role of ROS signaling during ER stress. Recent findings have shown that, unlike in metazoans, ROS produced by plant NADPH/respiratory burst oxidase protein-D (RBOHD) and -F (RBOHF) have a pro-survival role[58]. In contrast, $ire1a$ $ire1b$ mutant showed an increased $H_2O_2$ accumulation that may be associated with its accelerated death under ER stress[58]. Similar to these results, an $ire1$ knockout mutant in $Chlamydomonas$ $reinhardtii$ showed growth arrest and cytolysis under ER stress conditions associated with an increase in ROS marker gene expression under these conditions[72]. This dual role of ROS signaling and UPR activation has been supported by recent findings in $ch1$, a mutant of the isoprenoid biosynthesis pathway that produces more $^1O_2$ from PSII[73]. In this work, the authors showed that low production of singlet oxygen ($^1O_2$) caused a moderate UPR activation in $ch1$ leading to an acclimatory response. However, exposure to high light stress of $ch1$ induced a strong activation of the UPR that was associated with cell death. The results presented in our work support the notion that IRE1 is necessary to control cellular $H_2O_2$ concentrations under ER stress[58,72] and show that such control may be executed in coordination with BAP2, based on the decrease of $H_2O_2$ concentration observed in $ire1a$ $ire1b$ $bap2$. Similar to $ire1a$ $ire1b$, the induction of $H_2O_2$ in $bap2$ verified in our work may have a role in advancing PCD (Fig. 8).

The molecular mechanisms through which the plant UPR triggers PCD when ER stress cannot be resolved have begun to be understood. NAC089, a plant-specific ER-membrane anchored transcriptional factor from the Arabidopsis NAC family, has been identified as a positive regulator of ER stress-induced PCD by linking both the ER-stress sensing and downstream regulators of ER stress-induced cell death responses[34]. Under ER stress conditions, NAC089 relocates from the ER membrane to the Golgi, where it undergoes proteolytic cleavage by an unknown protease[74]. Once NAC089 is processed, it is redirected to the nucleus to induce the expression of PCD-associated genes. The NAC089 proteolytic activation also promotes an increase of caspase 3/7-like activity, ROS accumulation, and DNA fragmentation, and it is a tightly controlled process that takes place only under severe ER stress[34]. Therefore, NAC089 function can be regulated at both transcriptional and post-transcriptional levels. At a transcriptional level, $NAC089$ is directly induced by bZIP60 and bZIP28[34]. The results provided in our work have brought to light a pro-death role of BAP2 in controlling NAC089 expression under ER stress conditions, which attributes to BAP2 a dual role in PCD regulation. Together with the genetic evidence that BAP2 largely regulates $NAC089$ induction through the IRE1-bZIP60 arm, but not through bZIP28, these results support that IRE1 and BAP2 may induce $NAC089$ expression during the adaptive phase of the UPR as a preventive process to ignite PCD when the UPR becomes insufficient in unresolved ER stress, and $NAC089$ expression becomes mediated primarily by BAP2. Although our results suggest that a lower $H_2O_2$ accumulation and decreased $NAC089$ transcript levels may be associated with the less sensitive ER stress

phenotype of *ire1a ire1b bap2* compared to *ire1a ire1b*, they do not exclude the possibility that other factors may be involved in the BAP2 pro-death function and underscore a broader interconnection between cellular signaling pathways that warrants future investigation.

Genetic depletion of *BAP1* and *BAP2* together is lethal[35], and pathogen response analyses have attributed a role to BAP1 and BAP2 as negative regulators of PCD in biotic stress[35]. PCD is one of the many defense strategies that plants use to limit pathogen spread[75,76]. The evidence that, unlike *bap1*, *bap2* has no evident growth defects or increased disease resistance in response to pathogens, and that expression of *BAP2* rescues the *bap1* loss-of-function mutant phenotype[35] led to the suggestion that BAP1 and BAP2 may have partially overlapping functions in suppressing cell death in biotic stress[35]. The results presented in our work that *BAP2*, but not *BAP1* nor the BAP1-interactor *BON1* are required to overcome ER stress support the hypothesis that BAP2 and BAP1 have unique roles favoring either PCD or cell survival that are manifested in specific circumstances to regulate PCD in response to different stress cues. Therefore, based on the results presented in this work, we suggest that BAP2 may connect the UPR to other signaling pathways, including redox signaling, for cell fate determination.

## Methods

### Plant material and growth conditions

Seeds of the *A. thaliana* natural accessions (Supplementary Data 1) and the mutant lines *ire1a* (Col-0; WISCDSLOX420D09), *ire1b* (Col-0; SAIL_238_F07), *bzip28* (Col-0; SALK_132285), *bzip60* (Col-0; SALK_050203), *bap2* (Col-0; Salk_052789), and *bap1* (Col-0; SALK_092421) were obtained from the Arabidopsis Biological Resource Center (Columbus, OH, USA). The *bon1* mutant has been previously characterized[53,54]. *A. thaliana* ecotype Columbia-0 (Col-0) was used as a wild-type reference. Surface-sterilized seeds were plated directly on half-strength Linsmaier and Skoog (LS) medium, 1.0% w/v sucrose and 0.8% or 1.2% agar and then, were stratified in the dark at 4 °C for 2 days. Plates were transferred to a controlled growth chamber and incubated at 21 °C under continuous white light. Na-1 (CS1385) and Est-0 (CS6700) accessions of *A. thaliana* were crossed to obtain F$_1$. F$_1$ were selfed to produce 400 F2 plants which were selfed to generate 400 F$_3$ individuals to be used as the mapping population. The genotyping and isolation of multiple T-DNA insertion mutants were performed by genomic DNA extraction followed by DNA amplification using T-DNA- and gene-specific primers[18]. Primers are listed in Supplementary Data 5.

### Phenotypical analyses

For chronic ER stress assays, plants were germinated on 1/2 LS, 0.8% agar plates containing DMSO, 5, 10, 20, 25, or 50 ng mL$^{-1}$ TM, and growth for 10 days. Groups of 13 shoots were excised, and fresh weight was recorded. The shoot fresh weight ratio was calculated by dividing the fresh weight of TM-treated shoots by the fresh weight of DMSO (control)-shoots. Col-0 was grown on each plate as an internal control. At least three independent experiments were performed. Chlorophyll (chl) content quantification was performed as described previously[31]. Briefly, samples were ground to a fine powder in a Retch MM301 (Retch; Haan, Germany) by agitation at a frequency of 30 sec$^{-1}$ for two sets of 30 sec. Next, samples were incubated with 1 ml of DMSO for 20 min in the dark at room temperature. The samples were then centrifuged at 21,000 × *g*, and 200 µL of each supernatant was added to a clear 96-well plate for spectrophotometric quantification of chlorophyll content using a SpectramaxM2 (Molecular Devices). Finally, total chlorophyll content (chlorophyll a + chlorophyll b) per mg fresh weight was determined, and the relative value was calculated as (Chl$_{TM}$/Chl$_{DMSO}$). At least 12 biological replicates from 3 independent experiments were analyzed.

For temporary ER stress recovery assays, 5-days-old seedlings grown vertically on 1/2 LS, 1.2% agar plates were transferred to liquid 1/2 LS medium containing 1% sucrose and 0.5 mg mL$^{-1}$ TM or DMSO for 6 h, followed by transfer to 1/2 LS, 1.2% agar plates for 4 days. The root lengths of 72 seedlings from six different plates were measured using Image J. The relative root growth was calculated by measuring the root length after the TM treatment divided by the root length after the DMSO treatment. At least three independent experiments were performed.

Statistical analyses of both raw and relative responses (e.g., data relativized by DMSO control shoots) were conducted using linear mixed models in JMP Pro 17. Here, we considered the experiment and plate nested within the experiment as random factors and genotypes and ER stress treatments as fixed factors. Initial models tested for the main effect of genotype (e.g., mutants versus wild type), ER stress treatment (e.g., TM dose), or their interaction in a factorial model using an alpha threshold of 0.05[77]. To further explore treatment group differences, especially in models with significant genotype x treatment interaction, we split the datasets by TM dose and reanalyzed the data using a reduced model. Here, genotype means were contrasted at a given stress level using two-sided Tukey's HSD posthoc test with a multiple testing adjusted alpha threshold of 0.05 applied to the entire collection of pairwise comparisons. Data presented in the figures represent the least-squares mean ± SEM. The letters indicate which tests are different at multiple tests controlled by a 0.05 threshold.

### Accession analysis and QTL-seq mapping

*A. thaliana* accessions and the mapping population were grown on 1/2 LS, 0.8% agar plates containing DMSO or 50 ng mL$^{-1}$ TM for 10 days at which point fresh shoot weight data were obtained. Multiple accessions were grown in the same plate, including Col-0 which was grown in each plate as a reference control. These data were analyzed using a mixed linear model based on a factorial design, including accession, treatment, and their interaction as fixed effects and plate as a random effect using SAS Proc MIXED. The relative biomass ratio (RR) was calculated as

$$RR = [\text{Accession or } F_3\,(G_{50\,TM}/G_{DMSO})]/[Col - 0\,(G_{50\,TM}/G_{DMSO})] \quad (1)$$

where G is the fresh shoot weight and Col-0 is the values of Columbia-0 accession, which was used as reference accession. 10 mg of shoot tissue from each F$_3$ was collected in each replicate and stored at − 80 °C. The RR of 350 *A. thaliana* accessions and the 400 F$_3$ lines were analyzed in three independent biological replicates.

For the QTL mapping, the RR of the 400 F3 progenies was ranked, and the 10% tail of F3 with the lowest value was selected as the hypersensitive population, while the 10% F3 with the highest RR was selected as the hyper-resistant population. For both populations, 20 mg of shoot tissue from each selected F$_3$ was pooled together and homogenized using mortar and pestle. Three sets of 200 mg of the homogenized powder were collected in microcentrifuge tubes and were used to extract the genomic DNA using DNeasy Plant Mini Kit (Qiagen). The three samples of each population were combined together by equal amounts of DNA, and samples were sent to whole -genome sequencing using the Illumina HiSeq 4000 system.

The quality of the raw reads was assessed using the FastQC tool[78]. Sequencing adapters were trimmed from both pairs of raw reads using cutadapt v1.14[79]. Hyper-resistant (R) and hyper-sensitive (S) pools of filtered reads were aligned to the *A. thaliana* Columbia (Col-0) reference genome (source: TAIR10) using bwa mem algorithm[80] with default parameters. SNP-calling was performed by the mpileup function from bcftools[81]. A SNP-index, which is the proportion of SNPs that have different alleles other than the Col-0 reference alleles, was calculated for both R and S pool separately[82]. Next, we calculated a ΔSNP-index as the difference of the proportion of alternative allele between the two pools was calculated by subtracting SNP-index of R pool from the SNP-index of S pool. The average distributions of the SNP-index

and ΔSNP-index for a given genomic interval were estimated by using a sliding window approach with 1 Mb window size and 10 kb step. Confidence intervals were obtained by simulating an $F_3$ mapping population with the same pool size of this bulk segregating population ($n = 84$) with 10,000 replications for a given sequence depth. This process was replicated for a range of sequence depths (80–120) to obtain CI precisely for a given sequence depth. The SNP-index graphs for R-pool and S-pool, as well as the corresponding ΔSNP-index graph, were plotted using the ggplot2 package in R. Genomic intervals that crossed the 95% CI threshold of ΔSNP-index were considered as candidate genomic regions harboring a locus associated with ER stress.

## Plasmid construction and plant transformation

For complementation experiments, *BAP2* and its native promoter (1,002 bp upstream of translation start codon ATG and 624 bp of BAP2-coding region) were amplified from isolated Na-1 or Est-0 DNA. Primers used are listed in Supplementary Data 5. The initial PCR product was cloned into pCR8™/GW/TOPO and transferred to the pGWB4 (Gateway System, Invitrogen) destination vector. Constructs were introduced into *Agrobacterium tumefaciens* strain GV3101 and *A. thaliana* plants were transformed by a standard floral dip method[83]. For recombinant protein production, *BAP2* or *BAP2N67S* cDNA was cloned into pGEX5x-1 vector for GST-fusion while IRE1B cytosolic domains obtained from *IRE1B* cDNA was cloned into pET28b vector for His6 fusions at the C-terminus and *BON1* cDNA was cloned into pET16b vector for His6 fusions at the N-terminus using In-Fusion Cloning system (Takara). Primers were obtained using the In-Fusion Cloning Primer Design Tool (Takara). All constructs were confirmed by sequencing. All primer sequences for those constructs are listed in Supplementary Data 5. pET28b-VAP27cyto was obtained from Stefano et al.[63].

## RNA extraction and quantitative RT-qPCR analysis

5-days-old seedlings grown on 1/2 LS, 1.2% agar plates were transferred to liquid 1/2 LS medium containing 1% sucrose and 1.0 mg mL⁻¹ TM or DMSO for 3 h or 6 h. For chronic ER stress analyses, 5-days-old seedlings grown on 1/2 LS, 1.2% agar plates were transferred to 1/2 LS, 1.2% agar plates containing 1.0 mg mL⁻¹ TM or DMSO for 3, 6, 24, or 48 hours as indicated. Total RNA was extracted from whole seedlings using the Macherey-Nagel NucleoSpin RNA Plant Kit (www.mn-net.com) following the manufacturer's instructions. Total RNA was used for RT-PCR and qRT-PCR[18]. Briefly, RNA samples within the experiment were reverse transcribed using iScript Reverse Transcriptase (Bio-Rad). Next, real-time quantitative RT-PCR with SYBR Green detection was performed in triplicate using the Applied Biosystems 7500 fast real-time PCR system. Data were analyzed by the DDCT method. Primers used in this study are listed in Supplementary Data 5. Based on the 1001 Arabidopsis genome database[56], no differences were found in the primer sequences from Na-1, Est-0, and Col-0 accessions; however, we observed different expression levels in *UBQ10* control gene between Est-0 and Na-1 or Col-0 accessions. Data were normalized to the expression of UBQ10 except for analysis sets with Est-0, where GAPDH was used to normalize the transcript levels. For each biological replicate, the individual values represented are the mean of three technical replicates. At least three independent experiments were performed. Statistical significance was established with the Student's two-tailed unpaired *t* test, assuming equal variance.

## GST-BAP2 and IRE1B-His in vitro interaction assay

The in vitro interaction assay was conducted as described previously[84]. Briefly, GST-BAP2, GST-BAP2N67S, IRE1B-His, BON1-His and His-VAP27cyto were expressed in *E.coli* BL21. Then, the bacterial cultures were centrifuged at $4000 \times g$ for 10 min. Next, pellets were homogenized in an equilibration buffer (125 mM Tris-HCl, 150 mM sodium chloride; pH 8.0), and cells were disrupted by sonication (4 x 30 s,

12 W). Finally, protein extracts were obtained by centrifugation ($12,000 \times g$, 30 min, 4 °C). GST-BAP2, GST-BAP2N67S, and GST protein extracts were incubated with resin for GST binding (Pierce™ Glutathione Agarose, Thermo Scientific) for 30 min on ice. Upon resin binding, BON1-His or His-VAP27cyto were incubated with recombinant GST-BAP2, while IRE1B-His was incubated with GST-BAP2, GST-BAP2N67S, or GST. A sample from each protein extract used as input was obtained before the incubation with the resin for GST binding. Samples were loaded on a 10% SDS-page gel using the same amount of total protein. Western blot analysis was performed using an anti-His monoclonal antibody (Santa Cruz Biotechnology, Cat# sc-8036, lot# L0220) using a dilution of 1:2000 with an incubation time of 2 h. GST-BAP2 was detected using an anti-GST monoclonal antibody (Santa Cruz Biotechnology, Cat# sc-138, lot#B2124) using a dilution 1:3000 with an incubation of 2 h. The in vitro interaction assay was performed three times with consistent results.

## H₂O₂ accumulation and electrolyte leakage measurements

7-days-old seedlings grown on 1/2 LS, 1.2% agar plates were transferred to 1/2 LS, 1.2% agar plates containing 0.25 mg mL⁻¹, 0.5 mg mL⁻¹ and 1.0 mg mL⁻¹ TM or DMSO for 48 h or they were germinated on 1/2 LS, 0.8% agar plates containing DMSO, 5, 10, or 20 ng mL⁻¹ TM and grown for 10 days. Then, $H_2O_2$ quantification or electrolyte leakage measurement was performed as described previously[58]. Briefly, to perform $H_2O_2$ quantification, 30–60 mg of tissue were collected for each sample in microcentrifuge tubes, frozen using liquid $N_2$, and ground using glass beads. Then, 200 mL of 10% trichloroacetic acid (TCA) were added to frozen samples and samples were centrifugated at $21,000 \times g$ and 4 °C for 20 min. 150 mL of the supernatant were added to microcentrifuge tubes containing 150 mL of 1 M sodium bicarbonate for neutralization. 100 mL of the neutralized samples were stored on ice, and the remaining samples were treated with catalase. To do this, 10 mL of ammonium sulfate catalase suspension (Sigma, C3515-10MG) were centrifuged, and the pellet was resuspended with 600 mL ddH₂O. 5 mL of this catalase working solution were added to the remaining neutralized samples, which were incubated at room temperature for 10 min. Then, the catalase-treated samples were stored on ice. Finally, 25 mL of neutralized samples and catalase-treated samples were transferred into a 96-well plate where 75 mL of Amplex Ultra Red (AUR) assay solution (50 mM AUR; 5 mg mL⁻¹ commercial horseradish peroxidase) were added. Samples were incubated in the dark at room temperature for 5 min. Then, the microplates were read using a SpectramaxM2 (Molecular Devices) using excitation l = 544 nm, emission l = 590 nm.

For the electrolyte leakage measurements, groups of 5 seedlings were briefly washed in ddH₂O and incubated in 4 mL of ddH₂O for 3 h in 15 mL conical tubes with gentle agitation. Then, liquid conductivity was measured. Next, tubes containing the samples were autoclaved (20 min, liquid cycle) and allowed to cool under gentle agitation for 3 h. Finally, total conductivity was measured, and the percentage of electrolyte leakage was calculated by dividing total conductivity by liquid conductivity.

At least three independent experiments were performed, and statistical significance was determined using a factorial linear mixed model framework followed by post-hoc testing using Tukey's HSD test as described above.

## Trypan blue staining

Trypan blue staining of shoot tissues was adapted from Johansson et al.[59]. Briefly, 7-days-old seedlings grown on 1/2 LS, 1.2% agar plates were transferred to 1/2 LS, 1.2% agar plates containing 0.25 mg mL⁻¹, 0.5 mg mL⁻¹, and 1.0 mg mL⁻¹ TM or DMSO for 48 h. Then, they had the shoot tissues excised and incubated in a freshly prepared trypan staining solution (0.025% trypan blue in an equal volume of water, glycerol, phenol, and lactic acid). Samples were incubated in a 90 °C

heating block for 2.5 min and then incubated at room temperature for a further 30 min. The staining solution was removed and stained tissues were rinsed 3 times with sterile distilled water. Samples were then destained overnight in the choral hydrate alternative Visikol (Visikol, Inc.; Hampton, USA), then mounted in fresh Visikol on slides and imaged with the microscope function of an Olympus Tough F2.0 camera (Olympus; Tokyo, Japan) equipped with a light ring. The trypan blue staining was performed three times with consistent results.

## Reporting summary

Further information on research design is available in the Nature Portfolio Reporting Summary linked to this article.

## Data availability

All data supporting the findings of this study are available within this paper and its Supplementary Information files. Est-0 and Na-1 WGS data are deposited in the BioSample database and are accessible through the BioSample accession codes SAMN42008350 and SAMN42008351, respectively, which can be accessed using the following link and, respectively. QTL data supporting the finding of this study have been deposited in the BioSample database and are accessible through the BioProject accession code PRJNA1125890 which can be accessed using the following link. QTL analysis data are provided in Supplementary Data 3. All the biological materials generated in this work are available from the authors upon request. Source data are provided with this paper.

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

## Acknowledgements

This study was supported primarily by the National Institutes of Health (R35GM136637; GM101038), with contributing support from Chemical Sciences, Geoscience and Biosciences Division, Office of Basic Energy Sciences, Office of Science, U.S. Department of Energy (DE-FG02-91ER20021), the Great Lakes Bioenergy Research Center, U.S. Department of Energy, Office of Science, Office of Biological and Environmental Research (DE-SC0018409) and MSU AgBioResearch (MICL02598) to F.B.; and with contributing support from the Grant APOSTD/2020/199 funded by the Generalitat Valenciana and Grant IJC2020-045165-I funded by MCIN/AEI/ 10.13039/501100011033 and by "European Union NextGenerationEU/PRTR", and Grant PID2020-119111GB-I00 funded by MCIN/AEI/ 10.13039/501100011033 for partial salary and partial research support to N.P.C.

## Author contributions

N.P.C., T.E.J., and F.B. designed research; N.P.C, E.R.A, C.R., T.J., X.W., B.R., and T.H. performed the research; N.P.C., X.W., T.E.J., and F.B. interpreted the results; N.P.C., T.E.J., and F.B. wrote the manuscript.

## Competing interests

The authors declare no competing interests.
