## [Peer Review File · Nature Communications]

Programmed cell death regulator BAP2 is required for IRE1-mediated unfolded protein response in Arabidopsis.Reviewers' Comments:

Reviewer #1:

Remarks to the Author:

In plants there is a lot to learn about the way cell 'decide' between survival or death during chronic ER-stress induced by environmental conditions. This is a fundamental question in cell biology, on which we know very little. In addition, understanding that life/death switch has relevant applications to mitigate the effect of drought or salt stress on crops root systems. The originality of the Ms is that the authors turned to natural variation to identify components of the life/death switch and identified a key player.

This is a well written Ms, with experiments beautifully carried out and displayed.

Unfortunately, I disagree with the interpretation of the data in figure 6 and it seems to me that the overall conclusion is the opposite of the one written and presented in fig 7, for chronic stress. If I misunderstood the text or figure 6, I apologise and will be happy to read the authors' reply. I suggest, however, that the conclusions should be changed and the model in figure 7 redrawn. I suggest that the new conclusion might need one extra experiment to support the model.

Major points

1. In Figure 7, if the model was true: *bap-2* should die more but it dies less in fig6, and *ire* should die less, but it dies more in fig6. Figure 6 suggests Instead that BAP-2 is required for the pro-death effect of the *ire* mutation. Fig5 suggests that in the absence of IRE, the down-regulation of BAP-2 expression is lifted in the *ire* mutant. This suggests that BAP-2 induces death unless it is downregulated by IRE. Figure 4 suggests that the role of BAP2 might be more complicated as in high dose conditions *bap-2* dies a bit more but not as much as *ire* which has got the highest ion leakage. See sketch uploaded. IRE inhibits BAP2, BAP2 inhibits IRE, BAP2 induces PCD. if you remove IRE, BAP2 induce death, if you remove both IRE and BAP2 you have less death. in wild type, chronic Er-stress alters the balance of inhibitions towards death possibly based on the different half-life of the proteins
2. I realise that this goes against the conclusion of Yang et al 2007, that BAP2 inhibits PCD. It should be noted that in Yang et al, clear inhibition is obtained only when both BAP2 and BON1 are co-expressed. True that 35S-BAP2 does not induce PCD in Yang et al conditions, using agro-infiltration of *N.bent* leaves. In this Ms, a BAP-2 KO reduces growth and death (fig6). Expression of BAP-2 partners might be important for its function.
3. Fig4 and fig 6 have the opposite effect of *bap-2* on cell death, that might be due to the conditions (high dose versus low dose). Fig4 might benefit of having the *ire/bap-2* background tested to fully compare to fig6.
4. The major effect of BAP-2 seems to be on reducing growth before cell death occurs and BAP-2 might not directly regulate PCD, however it appears as a great candidate to be a component of the life/death switch.
5. I feel that the authors should try to investigate if BAP-2 is directly pro-death or not during ER stress in their experimental system. This could be done using 35S-BAP-2 or Dex-BAP-2 + Tm to show if it induces death or not. In WT and *ire* background.
6. Because AtNAC089 has been suggested to induce PCD during ER stress it might be interesting to measure AtNAC089 using the RNA samples of figure 5.

Comments:

7. Line 14 'We established that BAP2 expression is induced by ER-stress in an inositol-requiring enzyme 1 (IRE1)-dependent manner' this goes against figure 5a where BAP-2 is strongly induced in the absence of IRE
8. Line 58 IRE...'ignites pro-death processes when ER stress cannot be resolved' does not seem to be the case in this ms, where *ire* mutant dies more than WT
9. Figure 2, please indicate what -1 and -2 are in *Bap2-1* and *Bap2-2*. initially, it is confusing for the

reader as it reads like different alleles, but I guess it is referring to 2 independent lines; or use #1 and #2. see line 876: e)

10. Title of fig 2 and 3 are not clear in relation to the content of the figures. The term 'natural variation' is vague, would be better to use 'Na-1 or Est-0 allele'. In addition, it would be easier for the reader that the titles indicate clearly that one uses the bap2 background and the other is using the Est-o background

11. Figure 4 seedling in panel a don't look dead? Is ion leakage measured later than 48h? what does trypan blue suggest?

12. Figure 6 BAP2 antagonizes the pro-survival role of IRE1 at low-concentration of Tm. if this was true then bap2 would survive better. here when survival is compromised in ire the presence of BAP2 makes it worse as irexbap2 dies more. in ion leakage data, therefore, IRE antagonise the pro-death of BAP2, in shoot and chlorophyll, BAP2 is required for the negative effect of ire mutation, then here BAP2 is below IRE and partially required to go to PCD. Rather IRE antagonises the pro-death role of BAP-2. In other words, IRE survival role is dual: activate UPR and antagonise pro-death BAP

Reviewers' Comments:

Reviewer #1:

Remarks to the Author:

In plants there is a lot to learn about the way cell 'decide' between survival or death during chronic ER-stress induced by environmental conditions. This is a fundamental question in cell biology, on which we know very little. In addition, understanding that life/death switch has relevant applications to mitigate the effect of drought or salt stress on crops root systems. The originality of the Ms is that the authors turned to natural variation to identify components of the life/death switch and identified a key player.

This is a well written Ms, with experiments beautifully carried out and displayed.

Unfortunately, I disagree with the interpretation of the data in figure 6 and it seems to me that the overall conclusion is the opposite of the one written and presented in fig 7, for chronic stress. If I misunderstood the text or figure 6, I apologise and will be happy to read the authors' reply. I suggest, however, that the conclusions should be changed and the model in figure 7 redrawn. I suggest that the new conclusion might need one extra experiment to support the model.

Major points

1. In Figure 7, if the model was true: *bap-2* should die more but it dies less in fig6, and *ire* should die less, but it dies more in fig6. Figure 6 suggests Instead that BAP-2 is required for the pro-death effect of the *ire* mutation. Fig5 suggests that in the absence of IRE, the down-regulation of BAP-2 expression is lifted in the *ire* mutant. This suggests that BAP-2 induces death unless it is downregulated by IRE. Figure 4 suggests that the role of BAP2 might be more complicated as in high dose conditions *bap-2* dies a bit more but not as much as *ire* which has got the highest ion leakage. See sketch uploaded. IRE inhibits BAP2, BAP2 inhibits IRE, BAP2 induces PCD. if you remove IRE, BAP2 induce death, if you remove both IRE and BAP2 you have less death. in wild type, chronic Er-stress alters the balance of inhibitions towards death possibly based on the different half-life of the proteins
2. I realise that this goes against the conclusion of Yang et al 2007, that BAP2 inhibits PCD. It should be noted that in Yang et al, clear inhibition is obtained only when both BAP2 and BON1 are co-expressed. True that 35S-BAP2 does not induce PCD in Yang et al conditions, using agro-infiltration of *N.bent* leaves. In this Ms, a BAP-2 KO reduces growth and death (fig6). Expression of BAP-2 partners might be important for its function.
3. Fig4 and fig 6 have the opposite effect of *bap-2* on cell death, that might be due to the conditions (high dose versus low dose). Fig4 might benefit of having the *ire/bap-2* background tested to fully compare to fig6.
4. The major effect of BAP-2 seems to be on reducing growth before cell death occurs and BAP-2 might not directly regulate PCD, however it appears as a great candidate to be a component of the life/death switch.
5. I feel that the authors should try to investigate if BAP-2 is directly pro-death or not during ER stress in their experimental system. This could be done using 35S-BAP-2 or Dex-BAP-2 + Tm to show if it induces death or not. In WT and *ire* background.
6. Because AtNAC089 has been suggested to induce PCD during ER stress it might be interesting to measure AtNAC089 using the RNA samples of figure 5.

Comments:

7. Line 14 'We established that BAP2 expression is induced by ER-stress in an inositol-requiring enzyme 1 (IRE1)-dependent manner' this goes against figure 5a where BAP-2 is strongly induced in the absence of IRE
8. Line 58 IRE...'ignites pro-death processes when ER stress cannot be resolved' does not seem to be the case in this ms, where *ire* mutant dies more than WT
9. Figure 2, please indicate what -1 and -2 are in *Bap2-1* and *Bap2-2*. initially, it is confusing for the

reader as it reads like different alleles, but I guess it is referring to 2 independent lines; or use #1 and #2. see line 876: e)

10. Title of fig 2 and 3 are not clear in relation to the content of the figures. The term 'natural variation' is vague, would be better to use 'Na-1 or Est-0 allele'. In addition, it would be easier for the reader that the titles indicate clearly that one uses the *bap2* background and the other is using the Est-o background

11. Figure 4 seedling in panel a don't look dead? Is ion leakage measured later than 48h? what does trypan blue suggest?

12. Figure 6 BAP2 antagonizes the pro-survival role of IRE1 at low-concentration of Tm. if this was true then *bap2* would survive better. here when survival is compromised in *ire* the presence of BAP2 makes it worse as *irexbap2* dies more. in ion leakage data, therefore, IRE antagonise the pro-death of BAP2, in shoot and chlorophyll, BAP2 is required for the negative effect of *ire* mutation, then here BAP2 is below IRE and partially required to go to PCD. Rather IRE antagonises the pro-death role of BAP-2. In other words, IRE survival role is dual: activate UPR and antagonise pro-death BAPReviewer #2:

Remarks to the Author:

The authors identified BAP2 as a gene required for ER stress tolerance by using *Arabidopsis* natural genetic variation and quantitative trait locus analysis. The authors carried out further analyses and proposed that BAP2 functions upstream of the essential UPR master regulator IRE1 to control the amplitude of adaptive unfolded protein response (UPR) and the sufficiency of pro-survival UPR in chronic ER stress. The identification of the BAP2 locus for ER stress tolerance in plants is an interesting finding. However, because the data presented in the manuscript are limited to phenotypic and gene expression analyses in *ire1ab* and *bap2* mutants, they do not make a significant advance over the identification of BAP2. To support the authors' claim that BAP2 functions "as a new cellular rheostat that monitors the sufficiency of the plant UPR by controlling the role of IRE1(L.86-87)" and "BAP2 is necessary to control the activity of IRE1 (L.382-383)", protein level analysis such as interaction between BAP2 and IRE1 would be required.

Tm sensitive Est-0 harbors Ser instead of Asp at position 67 (L240, Fig.2c), and the authors introduced *Bap2*(Est-0) genomic fragments into *bap2* mutant in Col-0 background and claimed that Asp-to-Ser substitution accounts for Tm-sensitive phenotype. However, the experiments do not rule out the possibility that the BAP2 promoter is the cause. In other words, *Bap2* expression level rather than Ser/Asp difference of BAP2 protein accounts for different Tm sensitivity.

Why do the authors apply different Tm concentrations throughout the manuscript (e.g. 25 – 50 ng/mL in Figs.1b, 2, 3, 500 ng/mL in Fig. 1d, 1 ug/mL in Figs. 4 and 5)?

The authors state that BAP2 is induced by IRE1 (e.g. L310, L381), but this is misleading because BAP2 highly accumulates in *ire1a ire1b* mutant (Fig. 5a)

Reviewer #3:

Remarks to the Author:

Unfolded protein response (UPR) is elicited to restore ER homeostasis when unfolded or misfolded proteins are accumulated in ER. UPR could also lead to programmed cell death (PCD) when ER stress is unresolved. The current manuscript reports the natural variations in ER stress sensitivity in 350 *Arabidopsis* ecotypes and narrowed down the genomic region responsible for ER stress tolerance/sensitivity to about 70 candidate genes through QTL-seq by resequencing two segregated pools derived from two contrasting ecotypes Na-1 and Est-0. They focused on one of the candidate gene BAP2, a previous identified genes for PCD regulation. They have shown some evidences that

BAP2 is involved in ER stress responses, this work may add some interesting information on how ER stress is regulated in plants. However, there are some concerns to be addressed and additional experiments are needed to reach the correct conclusions.

1. The underlying molecular mechanisms for how BAP2 regulates ER stress response are not sufficiently explored in the paper. BAP1 and BON1 are involved in PCD as previously reported, but their mutants had no ER-stress phenotype, why BAP2 is so special in terms of regulating PCD under ER stress conditions?

2. Among the 70 candidate genes, because BAP2 is slightly upregulated by ER stress, it is chosen for further study. It is strongly suggested to discuss that other genes that are not regulated by ER stress at transcriptional level may also be responsible for the observed variations of ER stress sensitivity among different ecotypes.

3. For association study, it is ok to use relative biomass ratio for phenotypic analysis. However, for the phenotype comparison between genetic materials, it is important to show the data from the control (TM=0), it is not clear whether the plant biomass of each comparison is different under normal growth conditions, which is misleading.

4. In many cases, only one mutant line has been used, at least for BAP2 mutant analysis, two independent lines are required, especially when classical genetic complementation was not done.

5. BAP2 is more upregulated by ER stress in ire1a ire1b mutant, suggesting that IRE1 negatively regulates BAP2. In addition, the ire1a ire1b bap2 mutant phenotype is somehow similar to the ire1a ire1b mutant plants under ER stress condition, it is difficult to draw the conclusion that BAP2 functions upstream of the essential UPR master regulator IRE1 to control the amplitude of adaptive UPR.

6. Data from only one representative experiment was shown, which may weaken the paper.

6. Minor points:

Line 7, ?

Line 30, replace plant homologs with plant functional homologs, because the protein similarity between ATF6 and bZIP28 is rather low.

Line 39, recently IRE1C was identified in Arabidopsis by two independent groups.

Line 71, two proteins, SPL and BLI, are reported to regulate the activity of IRE1 in plants.

Line 505, if only technical replicates are shown in the paper, how to calculate SEM?

Figure 1, when comparing gene expression in different ecotypes, authors should explain if there are SNPs in primer binding regions among ecotypes.

Figure 5, BIP3 should be BiP3; uBZIP60 should be ubZIP60; Tm should be TM.

Reviewer #4:

Remarks to the Author:

Per agreement with the editor, I am only reviewing the pop genomics portion of this manuscript.

Here the authors use natural variation in Arabidopsis to identify a gene involved in the unfolded protein response (UPR). The first survey arabidopsis accessions to find those most resistant or most sensitive to tunicamycin treatment. The two extremes were crossed, and then bulk segregant analysis of F3 progeny was conducted. This allowed them to identify a region associated with variation in Tm sensitivity. qRT-PCR screening of genes in the candidate interval identified a strong candidate in BAP2. Subsequent transgenic rescue experiments with different alleles of BAP2 show that the two alleles do indeed have different activity and are responsible for at least part of the natural variation in Tm sensitivity.

I find the population genomics to be well done and convincing. Comments are minor:

Figures 1c and 1d: The asterisks indicating significance are missing. (Actually, same problem in Figures 2, 3, and 6).

Figure 1e The 10% cutoff lines for pooling seem to have been drawn incorrectly...there are more individuals in the high pool than the low pool, as drawn.

Figure 1g is missing.

We would like to thank the reviewers for their constructive criticism. We have endeavored to address all the points of the reviewers by editing the text and adding new experiments. Addressing the concerns experimentally required substantial efforts to generate and analyze high order mutants, provide new biochemical experiments, and introduce additional experimental procedures to follow cell death. We feel that the additional work has greatly improved the manuscript. Therefore, we are grateful for all the suggestions.

REVIEWER COMMENTS

Reviewer #1 (Remarks to the Author):

In plants there is a lot to learn about the way cell 'decide' between survival or death during chronic ER-stress induced by environmental conditions. This is a fundamental question in cell biology, on which we know very little. In addition, understanding that life/death switch has relevant applications to mitigate the effect of drought or salt stress on crops root systems. The originality of the Ms is that the authors turned to natural variation to identify components of the life/death switch and identified a key player. This is a well written Ms, with experiments beautifully carried out and displayed. Unfortunately, I disagree with the interpretation of the data in figure 6 and it seems to me that the overall conclusion is the opposite of the one written and presented in fig 7, for chronic stress. If I misunderstood the text or figure 6, I apologise and will be happy to read the authors' reply. I suggest, however, that the conclusions should be changed and the model in figure 7 redrawn. I suggest that the new conclusion might need one extra experiment to support the model.

We would like to thank this reviewer who went above and beyond by providing a sketch. As detailed below, we have performed additional experiments that allowed a clear interpretation of the results. We hope to continue the conversation on the model if necessary. Thank you.

Major points

1. In Figure 7, if the model was true: bap-2 should die more but it dies less in fig6, and ire should die less, but it dies more in fig6. Figure 6 suggests Instead that BAP-2 is required for the pro-death effect of the ire mutation. Fig5 suggests that in the absence of IRE, the down-regulation of BAP-2 expression is lifted in the ire mutant. This suggests that BAP-2 induces death unless it is downregulated by IRE. Figure 4 suggests that the role of BAP2 might be more complicated as in high dose conditions bap-2 dies a bit more but not as much as ire which has got the highest ion leakage. See sketch uploaded. IRE inhibits BAP2, BAP2 inhibits IRE, BAP2 induces PCD. if you remove IRE, BAP2 induce death, if you remove both IRE and BAP2 you have less death. in wild type, chronic Er-stress alters the balance of inhibitions towards death possibly based on the different half-life of the proteins

Response: We are grateful for the thoughtful comment. The addition of new data and consideration of the points that the reviewer raised have allowed us to revise the model.

Essentially, we propose a model where in ER stress conditions BAP2 would function as a rheostat to monitor IRE1 function, which, in turn, would control *BAP2* levels for an optimization of UPR sufficiency. However, when the constitutive pro-death processes determinate the cell fate, as in the *ire1a ire1b* mutant, BAP2 tilts the homeostatic balance between pro-life and constitutive pro-death pathways towards the latter, executing a pro-death function. Revised Figure 7 summarizes this model. Again, we are very grateful for the comments.

2. I realise that this goes against the conclusion of Yang et al 2007, that BAP2 inhibits PCD. It should be noted that in Yang et al, clear inhibition is obtained only when both BAP2 and BON1 are co-expressed. True that 35S-BAP2 does not induce PCD in Yang et al conditions, using agro-infiltration of *N.bent* leaves. In this Ms, a BAP-2 KO reduces growth and death (fig6). Expression of BAP-2 partners might be important for its function.

Response: We agree with the reviewer that the experimental set up of Yang et al. is different from ours and that Yang et al. used a transient overexpression in a heterologous system which may account for the differences. We also agree with the proposed hypothesis that expression of functional partner(s) may be necessary to modulate activity of BAP2. Therefore, overexpression of BAP2 may not be ideal to investigate the function of BAP2 in ER stress. Therefore, we have focused our work on a knockout line in *Arabidopsis*, which allowed us to investigate the functional relationship between BAP2 and the UPR master regulators. Our new results gathered in the *bap2* mutant, which has a decrease induction of the pro-death transcription factor *NAC089* compared to WT, have revealed that BAP2 has a role in promoting PCD, supporting that BAP2 has a dual role in controlling PCD signaling.

3. Fig4 and fig 6 have the opposite effect of *bap-2* on cell death, that might be due to the conditions (high dose versus low dose). Fig4 might benefit of having the *ire/bap-2* background tested to fully compare to fig6.

Response: We have added the requested experiments. This has allowed us to conclude that BAP2 has a pro-death function in mild chronic ER stress condition when IRE1 signaling is impaired.

4. The major effect of BAP-2 seems to be on reducing growth before cell death occurs and BAP-2 might not directly regulate PCD, however it appears as a great candidate to be a component of the life/death switch.

Response: We agree. Although Yang et al have reported that *bap1* has morphological defects and the double mutant *bap1 bap2* is lethal, we were not able to observe a significant difference in *bap2* mutant grown in mock compared to WT. To test if the phenotypical differences are due to differences in growth, we added shoot fresh weight measurements (Supplemental Figures 3, 18c and 19 a). These new results show that *bap2* mutant does not have morphological defects compared to WT.

5. I feel that the authors should try to investigate if BAP-2 is directly pro-death or not

during ER stress in their experimental system. This could be done using 35S-BAP-2 or Dex-BAP-2 + Tm to show if it induces death or not. In WT and *ire* background.

Response: We tried to get BAP2 overexpression lines with multiple rounds of transformation but failed. It is possible that the cell has mechanisms to avoid BAP2 overexpression. As discussed above, it is also possible that the generation of functional BAP2 overexpression lines may require adequate expression of a BAP2 interactor. We also feel that Yang et al. could not obtain BAP2 overexpression lines for their overexpression experiments with BAP2 because their experiments were performed by agro-infiltration of *N. benthamiana* leaves. We have added new data PCD measurement data and cell death data as well as NAC089 expression profiling in *bap2* mutant in combination with *ire1a ire1b*, *bzip60* and *bzip28* loss of function mutants. The analyses support a pro-death role of BAP2.

6. Because AtNAC089 has been suggested to induce PCD during ER stress it might be interesting to measure AtNAC089 using the RNA samples of figure 5.

Response: Thank you so much for this suggestion. Following your recommendation, we tested NAC089 transcription levels in *bap2* and *ire1a ire1b bap2* and we extended our investigation to *bzip60 bap2*, *bzip28 bap2* and *bzip28 bzip60 bap2* mutants that we generated. We found a pro-death role of BAP2 by controlling the induction of NAC89 under stress conditions, with occurs mainly through the IRE1-bZIP60 arm.

Comments:

7. Line 14 'We established that BAP2 expression is induced by ER-stress in an inositol-requiring enzyme 1 (IRE1)-dependent manner ' this goes against figure 5a where BAP-2 is strongly induced in the absence of IRE

Response: As we showed in Fig 1h; 5a, 5b, BAP2 is induced in Col-0 over a time course of ER stress. However, BAP2 transcription levels are highly upregulated in the absence of IRE1, suggesting that IRE1 controls BAP2 expression antagonistically. We have made changes in the main text to clarify it.

8. Line 58 IRE... 'ignites pro-death processes when ER stress cannot be resolved' does not seem to be the case in this ms, where *ire* mutant dies more than WT

Response: Thank you for highlighting this point. In WT, IRE1 has a biphasic role: it directs pro-life activities while monitoring endogenous pro-death processes when ER stress is still resolvable but leads pro-death processes when ER stress cannot be resolved. In ER stress, the *ire1a ire1b* mutant dies more quickly than WT as shown in this manuscript and in many other publications (Nagashima et al. 2011; Chen and Brandizzi, 2012; Mishiba et al., 2013). The evidence that a loss of the two Arabidopsis IRE1 paralogs accelerates cell death under chronic ER stress indicates that IRE1 is essential to the activation of pro-survival strategies to outweigh pro-death processes (Chen and Brandizzi, 2012; Mishiba et al., 2013).

9. Figure 2, please indicate what -1 and -2 are in Bap2-1 and Bap2-2. initially, it is confusing for the reader as it reads like different alleles, but I guess it is referring to 2 independent lines; or use #1 and #2. see line 876: e).

Response: Done

10. Title of fig 2 and 3 are not clear in relation to the content of the figures. The term 'natural variation' is vague, would be better to use 'Na-1 or Est-0 allele'. In addition, it would be easier for the reader that the titles indicate clearly that one uses the *bap2* background and the other is using the Est-o background

Response: Thank you for your suggestion. We have modified the term 'natural variation' to 'Na-1 or Est-0 allele' as appropriate to be more specific. In addition, we have modified both titles to indicate with background we were testing.

11. Figure 4 seedling in panel a don't look dead? Is ion leakage measured later than 48h? what does trypan blue suggest?

Response: Thank you for your suggestion. We have performed a Trypan blue staining assay (Supplemental Fig. 14) and found more extended trypan-blue positive areas in the TM-treated shoots of *bap2* and *ire1a ire1b* seedlings showed compared to Col-0.

12. Figure 6 BAP2 antagonizes the pro-survival role of IRE1 at low-concentration of Tm. if this was true then *bap2* would survive better. here when survival is compromised in *ire* the presence of BAP2 makes it worse as *irexbap2* dies more. in ion leakage data, therefore, IRE antagonise the pro-death of BAP2, in shoot and chlorophyll, BAP2 is required for the negative effect of *ire* mutation, then here BAP2 is below IRE and partially required to go to PCD. Rather IRE antagonises the pro-death role of BAP-2. In other words, IRE survival role is dual: activate UPR and antagonise pro-death BAP

Response: Thank you for your comments, the new data that we provide in this work lead us to propose that BAP2 has a pro-death function, at least by regulating *NAC089* expression. However, the decrease in *NAC089* transcript levels in *ire1a ire1b bap2* compared to *ire1a ire1b* (Fig. 6f-h) indicate that IRE1 and BAP2 have partially overlapping functions and support our model in which IRE1 functions in dependence of BAP2. These results are further supported by the decrease levels of H₂O₂ accumulation observed in *ire1a ire1b bap2* compared to *ire1a ire1b* (Fig 6d).

Reviewer #2 (Remarks to the Author):

The authors identified BAP2 as a gene required for ER stress tolerance by using Arabidopsis natural genetic variation and quantitative trait locus analysis. The authors carried out further analyses and proposed that BAP2 functions upstream of the essential UPR master regulator IRE1 to control the amplitude of adaptive unfolded protein response (UPR) and the sufficiency of pro-survival UPR in chronic ER stress. The identification of the BAP2 locus for ER stress tolerance in plants is an interesting finding. However, because the data presented in the manuscript are limited to phenotypic and

gene expression analyses in *ire1ab* and *bap2* mutants, they do not make a significant advance over the identification of BAP2. To support the authors' claim that BAP2 functions "as a new cellular rheostat that monitors the sufficiency of the plant UPR by controlling the role of IRE1(L.86-87)" and "BAP2 is necessary to control the activity of IRE1 (L.382-383)", protein level analysis such as interaction between BAP2 and IRE1 would be required.

Response: Thank you for your insightful comments. We agree. We performed the transcriptional analyses of *IRE1A* and *IRE1B* in the *bap2* mutant and as we show in Supplemental Fig. 17, there are no significant differences compared to WT, suggesting a direct interaction of IRE1 and BAP2. Therefore, we performed an *in vitro* interaction assay, which is included in Fig. 5g. The results show an *in vitro* association between BAP2 and IRE1. The evidence provided in the revised manuscript that BAP2 interferes with the splicing of bZIP60 mRNA suggests that this direct interaction may modulate the splicing activity of IRE1.

Tm sensitive Est-0 harbors Ser instead of Asp at position 67 (L240, Fig.2c), and the authors introduced Bap2(Est-0) genomic fragments into *bap2* mutant in Col-0 background and claimed that Asp-to-Ser substitution accounts for Tm-sensitive phenotype. However, the experiments do not rule out the possibility that the BAP2 promoter is the cause. In other words, Bap2 expression level rather than Ser/Asp difference of BAP2 protein accounts for different Tm sensitivity.

Response: This is a good point that we carefully considered. We have made several observations that have led us to exclude that possibility. First, as we show in Supplemental Fig. 12c, there is no difference in *BAP2* transcription levels between the accessions. Second, a comparison of the promoter regions across the accession shows a high degree of identity. In particular, the Est-0 promoter differs only in 1 nucleotide compared with Col-0 and Na-1 promoters, which are otherwise identical as we show in Supplemental Fig. 11. In support of these points, the functional results observed in the transgenic lines of both *bap2* mutant and Est-0 ecotype did not show significant differences in *BAP2* transcription levels. Hence we can exclude that BAP2 promoter is the cause of the observed phenotype. Again, we thank the reviewer for raising this very important point as it allowed us to highlight it in the manuscript.

Why do the authors apply different Tm concentrations throughout the manuscript (e.g. 25 – 50 ng/mL in Figs.1b, 2, 3, 500 ng/mL in Fig. 1d, 1 ug/mL in Figs. 4 and 5)?

Response: For chronic ER stress analysis our routine screen protocol uses 25 and 50ng/mL TM. Because low doses allow revealing subtle phenotype that could be otherwise missed by high dose of a drug and facilitate identifying proteins that can be key during ER stress cell determination in mild ER stress conditions, we extended our range from 15 to 50 ng/mL. Furthermore, the phenotype observed in *ire1a ire1b bap2* made us decrease our range to 5 to 20 ng/mL in Fig. 6.

For pulse experiments and prolonged ER stress analysis, we used the established concentration for these type experiments (1 ug/mL) to be consistent with the methodologies used in publications by our lab and others.

The authors state that BAP2 is induced by IRE1 (e.g. L310, L381), but this is misleading because BAP2 highly accumulates in *ire1a ire1b* mutant (Fig. 5a)

Response: As we showed in Fig 1h; 5a, 5b, *BAP2* is induced in Col-0 over a time course of ER stress. However, *BAP2* transcription levels are highly upregulated in absence of IRE1, suggesting that IRE1 controls *BAP2* expression antagonistically. We have made changes to the main text to clarify this point.

Reviewer #3 (Remarks to the Author):

Unfolded protein response (UPR) is elicited to restore ER homeostasis when unfolded or misfolded proteins are accumulated in ER. UPR could also lead to programmed cell death (PCD) when ER stress is unresolved. The current manuscript reports the natural variations in ER stress sensitivity in 350 Arabidopsis ecotypes and narrowed down the genomic region responsible for ER stress tolerance/sensitivity to about 70 candidate genes through QTL-seq by resequencing two segregated pools derived from two contrasting ecotypes Na-1 and Est-0. They focused on one of the candidate gene *BAP2*, a previous identified genes for PCD regulation. They have shown some evidences that *BAP2* is involved in ER stress responses, this work may add some interesting information on how ER stress is regulated in plants. However, there are some concerns to be addressed and additional experiments are needed to reach the correct conclusions.

1. The underlying molecular mechanisms for how *BAP2* regulates ER stress response are not sufficiently explored in the paper. *BAP1* and *BON1* are involved in PCD as previously reported, but their mutants had no ER-stress phenotype, why *BAP2* is so special in terms of regulating PCD under ER stress conditions?

Response: To address this point, we performed a transcriptional analysis of *IRE1A* and *IRE1B* in *bap2* mutant and, as we show in Supplemental Fig. 17, there are no significant differences compared to WT, suggesting a direct interaction of IRE1 and *BAP2*. Therefore, we have performed an *in vitro* interaction assay, which is included in Fig. 5g. The results show an interaction between recombinant *BAP2* and IRE1. Based on additional results provided in the revised manuscript on the levels of splicing of *bZIP60* mRNA, we propose that *BAP2* might control IRE1 splicing function via a direct protein-protein interaction similarly to the effect of BI-1 on metazoan IRE1.

2. Among the 70 candidate genes, because *BAP2* is slightly upregulated by ER stress, it is chosen for further study. It is strongly suggested to discuss that other genes that are not regulated by ER stress at transcriptional level may also be responsible for the observed variations of ER stress sensitivity among different ecotypes.

Response: Thank you for your suggestion. We have added a paragraph in the discussion section to discuss this point.

3. For association study, it is ok to use relative biomass ratio for phenotypic analysis. However, for the phenotype comparison between genetic materials, it is important to show the data from the control (TM=0), it is not clear whether the plant biomass of each comparison is different under normal growth conditions, which is misleading.

Response: Thank you for your suggestion. To show that the phenotypical differences are not due to differences in growth, we added shoot fresh weight measurements (Supplemental Figures 3 and 19 a) in all our phenotypical analysis. The analyses also include chlorophyll measurements (Supplemental Fig. 5).

4. In many cases, only one mutant line has been used, at least for BAP2 mutant analysis, two independent lines are required, especially when classical genetic complementation was not done.

Response: Thank you for your suggestion. We have performed the analyses with two independent transgenic lines. We used one *bap2* allele because we could complement it with the Na-1 BAP2 sequence. Specifically, we did not perform a genetic complementation with Col-0 sequence because Na-1 and Col-0 sequences only differ by one nucleotide which leads to a synonymous mutation (Supplemental Figs. 10 and 11). Therefore, the *pBAP2_{Na-1}:BAP2* in *bap2* background can be considered the equivalent of a complementation with the endogenous gene in *bap2*. As we show in Fig. 2b, *pBAP2_{Na-1}:BAP2* expression in *bap2* background complements *bap2* phenotype under ER stress. While these results justify the use of one *bap2* allele we note in the manuscript that the *bap2* allele used in our work has been established by Yang et al. (Plant Physiology, 2007).

5. BAP2 is more upregulated by ER stress in *ire1a ire1b* mutant, suggesting that IRE1 negatively regulates BAP2. In addition, the *ire1a ire1b bap2* mutant phenotype is somehow similar to the *ire1a ire1b* mutant plants under ER stress condition, it is difficult to draw the conclusion that BAP2 functions upstream of the essential UPR master regulator IRE1 to control the amplitude of adaptive UPR.

Response: Thank you for your comments, the addition of new data has allowed us to revise the model and support our hypothesis that BAP2 functions to influence the IRE1's function. Gene expression results, results obtained by monitoring cell death and ROS levels and biochemical analyses lend support to the model that BAP2 functions to control IRE1 and IRE1 controls BAP2 expression as a negative feedback loop.

6. Data from only one representative experiment was shown, which may weaken the paper.

Response: Thank you for your comment. In the revised manuscript, we have added the data from all the biological replicates that we performed in the three independent experiments. This led us to revise our statistical approaches for the phenotypical assays as explained in the revised Material and methods section.

6. Minor points:

Line 7, <designed> ?

Response: Edited to “has evolved”.

Line 30, replace plant homologs with plant functional homologs, because the protein similarity between ATF6 and bZIP28 is rather low.

Response: Thank you for your suggestion. Done.

Line 39, recently IRE1C was identified in Arabidopsis by two independent groups.

Response: Agree, but a function of IRE1C in splicing bZIP60 has not been reported yet. Therefore, we prefer to focus the introduction section on IRE1A and IRE1B, which have been demonstrated to be functional IRE1 paralogs. In the introduction we write “functional” to address this point.

Line 71, two proteins, SPL and BL1, are reported to regulate the activity of IRE1 in plants.

Response: Thank you for your comment. SPL and BL1 have been reported to regulate IRE1 activity under normal growth conditions, but not under ER stress conditions. In addition, SPL controls IRE1 function at transcriptional level while for BL1 is not clear. Although they have been proposed to regulate IRE1 in a different context, we added them in the discussion.

Line 505, if only technical replicates are shown in the paper, how to calculate SEM?

Response: Thank you for your comment. We realize that there is a misunderstanding. We have presented biological replicates. However, for qRT-PCR analyses and ROS assays, the values used in each biological replicate are the mean from three technical replicates. We have clarified this point in the Material and methods section.

Figure 1, when comparing gene expression in different ecotypes, authors should explain if there are SNPs in primer binding regions among ecotypes.

Response: Thank you for your suggestion. There are no SNPs in the primer binding region. We have indicated this in the figure legend and in the Material and methods section.

Figure 5, BIP3 should be BiP3; uBZIP60 should be ubZIP60; Tm should be TM.

Response: Changed according to the suggestions.

Reviewer #4 (Remarks to the Author):

Per agreement with the editor, I am only reviewing the pop genomics portion of this manuscript.

Here the authors use natural variation in Arabidopsis to identify a gene involved in the unfolded protein response (UPR). The first survey arabidopsis accessions to find those most resistant or most sensitive to tunicamycin treatment. The two extremes were crossed, and then bulk segregant analysis of F3 progeny was conducted. This allowed them to identify a region associated with variation in Tm sensitivity. qRT-PCR screening of genes in the candidate interval identified a strong candidate in BAP2. Subsequent transgenic rescue experiments with different alleles of BAP2 show that the two alleles do indeed have different activity and are responsible for at least part of the natural variation in Tm sensitivity.

I find the population genomics to be well done and convincing. Comments are minor:

Figures 1c and 1d: The asterisks indicating significance are missing. (Actually, same problem in Figures 2, 3, and 6).

Response: That you for your observation, we have added them.

Figure 1e The 10% cutoff lines for pooling seem to have been drawn incorrectly...there are more individuals in the high pool than the low pool, as drawn.

Response: Thank you for your observation. We have placed the green cutoff line at the correct position.

Figure 1g is missing.

Response: We have included it.

Reviewers' Comments:

Reviewer #1:

Remarks to the Author:

This revised manuscript now constitutes a strong contribution to our understanding of how plant cells 'decide' between survival or death during chronic ER stress induced by environmental conditions. This is a fundamental question in cell biology, which has relevant applications to mitigate the effect of drought or salt stress on crop root systems.

This is a well-written Ms, with experiments well carried out and high-quality displays of the results. For this revision, the authors have carried out many additional crosses and experiments to address successfully the reviewer's comments.

Minor points

376 and 380: we co-expressed or co-expression? It seems there was no co-expression in the same E.coli? Instead, each interactor was expressed in E.coli and combined in vitro for pull-down assays. Combining is a better description than co-expressing, which suggests expression in the same cell, in N. bent., for example.

405 estimated the ratio of shoot fresh weight and chlorophyll content? Why is it estimated rather than measured? if an estimation is used, it should be described in the method section

663 typo: Col-0 was grown

761 qRT-PCR as described previously. Ref needed

Fig 5g. It is a pity that the signal for the IRE pull-down could be more convincing. Currently, it looks like a dot, a random dot?, rather than a band. Do you have another blot or another exposure to add to the supplement?

Fig 7 could distinguish direct known effect as continuous arrows (___) eg IRE1 to Zip60 to BIP3, from presumably indirect (multi-step) effect with dotted arrows (- - - -) eg BAP2 to PCD . Zip 60 could be on the outside and go directly (___) to NAC089

Ref 64 to correct : proteins in the *Arabidopsis* unfolded protein response

—

Reviewer #2:

Remarks to the Author:

Regarding the authors' responses to Reviewer #2

The authors carried out in vitro pull-down assay in Fig. 5g to see if BAP2 could directly affect IRE1 at a protein level in response to the first concern to the previous version of the manuscript. It indeed shows the direct binding of BAP2 to IRE1 as expected, but needs further validation to support the direct role of BAP2 on IRE1. If the 67th amino acid (N in Na-1 and S in Est-0) accounts for the ability of BAP2 to control cell fate determination during ER stress, BAP2(N67) and BAP2(S67) would be expected to show different binding affinity to IRE1, which needs to be tested in a pull-down assay. Also, the Fig. 5g data are not of sufficient quality. The signal intensity of GST-BAP2 is uneven among lanes but should be nearly equal if the same amount of GST-BAP2 is used as a bait for different His-tagged prey proteins. The data should also include input lanes to show the initial amounts of His-tagged proteins used for the binding assay. In addition, the term "co-expressed" (L378-380) is not appropriate because the proteins for pull-down assays must be expressed individually.

Other concerns that were raised by Reviewer #2 have been addressed in the revised manuscript.

Regarding the authors' responses to Reviewer #3

The authors performed pull-down assays in Fig. 5g in response to the first concern raised by Reviewer #3, but it requires additional experiments as mentioned above.

The other concerns that were raised by Reviewer #3 have been addressed in the revised manuscript.

Reviewer #4:

Remarks to the Author:

The authors have addressed my previous concerns.

We would like to thank the reviewer for the positive comments and encouragement about the findings in our work. We have endeavored to address all the points raised for the resubmission but editing the text and adding the requested control. We feel that the revisions have helped support our original findings. Thank you again for the time and efforts for the work.

REVIEWER COMMENTS

Reviewer #1 (Remarks to the Author):

This revised manuscript now constitutes a strong contribution to our understanding of how plant cells 'decide' between survival or death during chronic ER stress induced by environmental conditions. This is a fundamental question in cell biology, which has relevant applications to mitigate the effect of drought or salt stress on crop root systems. This is a well-written Ms, with experiments well carried out and high-quality displays of the results. For this revision, the authors have carried out many additional crosses and experiments to address successfully the reviewer's comments.

Minor points:

376 and 380: we co-expressed or co-expression? It seems there was no co-expression in the same E.coli? Instead, each interactor was expressed in E.coli and combined in vitro for pull-down assays. Combining is a better description than co-expressing, which suggests expression in the same cell, in N. bent., for example.

Thank you for your suggestion. We have changed the term co-expression for combination.

405 estimated the ratio of shoot fresh weight and chlorophyll content? Why is it estimated rather than measured? if an estimation is used, it should be described in the method section

Thank you for your observation. We mean "calculated based on measurements", so we changed to "measured".

663 typo: Col-0 was growthn
Amended.

761 qRT-PCR as described previously. Ref needed
Thank you. We have added the missed reference.

Fig 5g. It is a pity that the signal for the IRE pull-down could be more convincing. Currently, it looks like a dot, a random dot?, rather than a band. Do you have another blot or another exposure to add to the supplement?
Since we performed a new in vitro interaction assay to include GST-BAP2N67S, we have changed the blot for a new one including IRE1B, which is now clearly visible.

Fig 7 could distinguish direct known effect as continuous arrows (___) eg IRE1

to Zip60 to BIP3, from presumably indirect (multi-step) effect with dotted arrows (- - - -) eg BAP2 to PCD . Zip 60 could be on the outside and go directly (____) to NAC089

Thank you for your suggestion to improve our model. We have made the necessary changes in order to distinguish between direct and indirect effects as requested.

Ref 64 to correct : proteins in the *Arabidopsis* unfolded protein response
Thank you, we have changed the reference.

—

Reviewer #2 (Remarks to the Author):

Regarding the authors' responses to Reviewer #2

The authors carried out in vitro pull-down assay in Fig. 5g to see if BAP2 could directly affect IRE1 at a protein level in response to the first concern to the previous version of the manuscript. It indeed shows the direct binding of BAP2 to IRE1 as expected, but needs further validation to support the direct role of BAP2 on IRE1. If the 67th amino acid (N in Na-1 and S in Est-0) accounts for the ability of BAP2 to control cell fate determination during ER stress, BAP2(N67) and BAP2(S67) would be expected to show different binding affinity to IRE1, which needs be tested in a pull-down assay. Also, the Fig. 5g data are not of sufficient quality. The signal intensity of GST-BAP2 is uneven among lanes but should be nearly equal if the same amount of GST-BAP2 is used as a bait for different His-tagged prey proteins. The data should also include input lanes to show the initial amounts of His-tagged proteins used for the binding assay. In addition, the term "co-expressed" (L378-380) is not appropriate because the proteins for pull-down assays must be expressed individually.

Thank you for your suggestion. We have performed a new in vitro interaction assay with the mutated version of BAP2, BAP2N67S fused to GST as we had done with BAP2-GST. As you can see in the new Fig. 5g, we didn't see an interaction between GST-BAP2N67S and IRE1Bcyt-His. We are grateful for the comment because it provided us the opportunity to explain that the plant phenotype is most likely due to the loss of interaction of BAP2 with IRE1B due to the N67S mutation. In the blot that we have used, the signal intensity of GST-BAP2 is quite similar in all the GST-BAP2 bands, except that for the interaction of GST-BAP2 with His-BON1 which is the positive control. Although we first incubated GST-BAP2 with the GST resin, then we split the same volume of the resin into in the corresponding number of samples and we checked it after centrifugation (in order to be as nearly equal as possible), we got different signal intensities between GST-BAP2 bands in the blots. However, we have been able to get similar signal intensity between GST-BAP2 and GST-BAP2N67S in the in vitro interaction blot as well as in the input blot. Input lines have been shown in Supplemental Fig. 18. As you can see in Supplemental Fig. 18b, IRE1Bcyt-His expression is lower compared to His-BON1 and His-VPA27cyt; however, we were able to see an interaction of IRE1B with GST-BAP2. IRE1Bcyt-His input is the same for GST-BAP2 and GST-BAP2N67S interaction assays since we equally

split the IRE1Bcyt-His pool for the interaction with GST-BAP2 and GST-BAP2N67S. Hence, the absence of an interaction of IRE1B with BAP2N67S is due to the residue mutation rather than different protein loading in the pull-downs. The term “co-expression” has been changed to combined as Reviewer #1 suggested.

Other concerns that were raised by Reviewer #2 have been addressed in the revised manuscript.

Regarding the authors' responses to Reviewer #3

The authors performed pull-down assays in Fig. 5g in response to the first concern raised by Reviewer #3, but it requires additional experiments as mentioned above.

Thank you for your comment. We have performed new in vitro interaction assays adding GST-BAP2N67S bait (Fig. 5g) and we have included the input information in the Supplemental Figure 18, as the previous reviewers requested.

The other concerns that were raised by Reviewer #3 have been addressed in the revised manuscript.

Reviewer #4 (Remarks to the Author):

The authors have addressed my previous concerns.
Thank you.

Reviewers' Comments:

Reviewer #1:

Remarks to the Author:

The authors have addressed my concerns and comments

Reviewer #2:

Remarks to the Author:

The current version of the manuscript has addressed the reviewer's previous comment.